

# Ice nucleating particles in the Saharan Air Layer

Yvonne Boose[1], Berko Sierau[1], M. Isabel García[2,3], Sergio Rodríguez[2], Andrés Alastuey[4],
Claudia Linke[5], Martin Schnaiter[5], Piotr Kupiszewski[6], Zamin A. Kanji[1], and Ulrike Lohmann[1]

[1]Institute for Atmospheric and Climate Science, ETH Zürich, 8092 Zürich, Switzerland
[2]Izaña Atmospheric Research Center, AEMET, Santa Cruz de Tenerife, Tenerife, Spain
[3]Department of Chemistry (T.U. Analytical Chemistry), Faculty of Science, University of La Laguna, Tenerife, Spain
[4]Institute of Environmental Assessment and Water Research, CSIC, Barcelona, Spain
[5]Institute for Meteorology and Climate Research, Atmospheric Aerosol Research, Karlsruhe Institute of Technology, Karlsruhe, Germany
[6]Laboratory of Atmospheric Chemistry, Paul Scherrer Institute, Villigen, Switzerland

*Correspondence to:* Y.Boose (yvonne.boose@env.ethz.ch) and Z. A. Kanji (zamin.kanji@env.ethz.ch)

**Abstract.** This study aims at quantifying the ice nucleation properties of desert dust in the Saharan Air Layer (SAL), the warm, dry and dust-laden layer that expands from North Africa to the Americas. By measuring close to the dust's emission source, before aging processes during the transatlantic advection potentially modify the dust properties, the study fills a gap between in-situ measurements of dust ice nucleating particles (INPs) far away from the Sahara and laboratory studies of ground-collected

soil. Two months of online INP concentration measurements are presented, which were part of the two CALIMA campaigns at the Izaña observatory in Tenerife, Spain (2373 m. asl.) in the summers of 2013 and 2014. INP concentrations were measured in the deposition and condensation mode at temperatures between 233 K and 253 K with the Portable Ice Nucleation Chamber (PINC). Additional aerosol information such as bulk chemical composition, concentration of fluorescent biological particles as well as the particle size distribution was used to investigate observed variations in the INP concentration.

The concentration of INPs was found to range between 0.2 stdl$^{-1}$ in the deposition mode and up to 2500 stdl$^{-1}$ in the condensation mode at 241 K. It correlates well with the abundance of aluminum, iron, magnesium and manganese (R: 0.43-0.67) and less with that of calcium, sodium or carbonate. These observations are consistent with earlier results from laboratory studies which showed a higher ice nucleation efficiency of certain feldspar and clay minerals compared to other types of mineral dust. We find that an increase of ammonium sulfate, linked to anthropogenic emissions in upwind distant anthropogenic sources,

mixed with the desert dust, has a small positive effect on the INP per dust mass ratio. Furthermore, the relative abundance of biological particles was found to be significantly higher in INPs compared to the ambient aerosol. Two common parameterization schemes for INP concentrations, which were derived mostly from atmospheric measurements far away from the Sahara, were found to predict more INPs based on the aerosol load than we observed in the SAL. Overall, this suggests that atmospheric aging processes in the SAL can lead to an increase in ice nucleation efficiency of mineral dust from the Sahara.





## 1 Introduction

About 65 % of the global desert dust emissions stem from North Africa (Ginoux et al., 2004). Saharan dust influences the Earth's radiative budget directly through scattering and absorption of solar radiation (Haywood et al., 2003). Dust particles may also act as cloud condensation nuclei (CCN) or ice nucleating particles (INPs), thus affecting cloud properties and con-

tributing to a radiative forcing due to aerosol-cloud interactions (Boucher et al., 2013). The uncertainties in quantifying these effects remain significant. Especially the ice phase has a major impact on cloud properties by influencing cloud lifetime and precipitation (Lohmann and Feichter, 2005; Boucher et al., 2013). Furthermore, warm, liquid clouds generally lead to a negative radiative forcing (cooling effect), whereas cirrus clouds potentially lead to a positive radiative forcing and thus may warm the climate.

Several laboratory studies during the last six decades have indicated the potent role of mineral dust as INP at temperatures below 263 K (Isono and Ikebe, 1960; Pruppacher and Klett, 1997; Hoose and Möhler, 2012; Murray et al., 2012) with certain feldspar minerals having the highest ice nucleating potential amongst the main mineral dust components (Atkinson et al., 2013; Yakobi-Hancock et al., 2013; Harrison et al., 2016). Also in the atmosphere, mineral dust has been observed to commonly be involved in ice nucleation in mixed-phase and cirrus clouds (DeMott et al., 2003; Cozic et al., 2008; Seifert et al., 2010; Cziczo

et al., 2013; Creamean et al., 2013). In some case studies it has been shown that mineral dust is dominating ice nucleation over Europe even outside of periods of high Saharan dust abundance (Klein et al., 2010; Chou et al., 2011; Boose et al., 2016). However, climatological estimates of dust INP concentrations are still missing (Hande et al., 2015).

30-50 % of the total emitted Saharan dust is transported westward in the Saharan Air Layer (SAL), making it the main global dust transport pathway (Carlson and Prospero, 1972; Goudie and Middleton, 2001). The SAL can be identified throughout the

20  year (Tsamalis et al., 2013). It follows a clear seasonal cycle related to the general circulation pattern. Peak dust emissions in West Africa are found in summer and are correlated with the northward shift of the Inter-Tropical Convergence Zone (Engelstaedter and Washington, 2007). The shift leads to increased surface gustiness in West Africa as well as dry convection and stronger vertical winds which results in an enhanced uplift of dust particles. The African easterly jet then forces the dry, dust-laden warm air to move westwards in the SAL at 600-800 hPa above the moist trade wind inversion (Carlson and Pros-

pero, 1972; Chiapello et al., 1995). In July and August, a maximum in number and intensity of dust events is reported for the Izaña Atmospheric Observatory in Tenerife which is frequently located within the SAL as reported by Rodríguez et al. (2015). The authors identified regions in the subtropical Sahara, a stripe expanding from central Algeria to northern Mauritania and Western Sahara, as main sources of dust advected to Izaña during the summer.

Other Saharan dust transport pathways are: from the Sahara northwards over the Mediterranean towards Europe (Collaud Coen

et al., 2004; Ansmann et al., 2005); toward the eastern Mediterranean, Middle East (Kubilay et al., 2000; Galvin, 2012) and as far as East Asia (Tanaka et al., 2005) or California (Creamean et al., 2013); and south, towards the Gulf of Guinea (Breuning-Madsen and Awadzi, 2005).

Dall'Osto et al. (2010) found that dust particles collected from the soil surface in the Sahara were hardly mixed with nitrate or sulfate. After being advected to Cape Verde dust particles were increasingly internally mixed with nitrate but not with sulfate.





When sampled at a coastal station in Ireland, the Saharan dust particles showed a very high degree of mixing with nitrate and sulfate. Kandler et al. (2007), on the other hand, used scanning electron microscopy of aerosol samples collected with a cascade impactor at the Izaña observatory and found that submicron mineral dust was coated with sulfate. Rodríguez et al. (2011) analyzed the bulk chemical composition of aerosol particles in the Saharan Air Layer collected over six years. Their

study showed that desert dust collected at Izaña is often mixed with nitrate, sulfate and ammonium as well as phosphorous originating from industrial emissions on the North African coast. Hence, the different transport pathways lead to different degrees of mixing of the dust aerosol. Knippertz and Stuut (2014) thus distinguish between "mineral dust", describing only those inorganic mineral particles originating from the soil and "desert aerosol", meaning all airborne particulates found in the outflow of the dust source.

Apart from being mixed with pollutants, the dust may undergo in-cloud or photo processing. A range of laboratory studies have shown that the ice nucleation ability of mineral dust particles can be altered by aging processes. Condensation of sulfuric acid (Knopf and Koop, 2006; Sihvonen et al., 2014; Wex et al., 2014) was observed to mostly impair ice nucleation, whereas ammonium (Salam et al., 2007; Koop and Zobrist, 2009), nitric acid (Sullivan et al., 2010), or the exposure to ozone (Kanji et al., 2013) can promote it. Biological material, which is mixed with the dust particles already in the soil or gets mixed during

the atmospheric transport, may also affect the ice nucleating behavior of the dust (Schnell and Vali, 1976; Michaud et al., 2014). Some biological particles, like the bacterium *Pseudomonas syringae*, have been observed to lead to ice nucleation at temperatures warmer than 258 K (see Hoose and Möhler 2012 and references therein). The importance of these different atmospheric processes is highlighted by observations of clouds over Florida glaciating at temperatures above 264 K during the presence of Saharan dust (Sassen et al., 2003) which is above the ice nucleation onset temperatures found in laboratory studies

for pure mineral dust (Hoose and Möhler, 2012; Murray et al., 2012). Conen et al. (2015) found a weak influence of Saharan dust events (SDEs) on the INP concentrations at 265 K at the Jungfraujoch in the Swiss Alps but an order of magnitude lower INP concentrations during SDEs at Izaña, suggesting that atmospheric processes led to enhanced ice nucleation ability of the Saharan dust after long-range transport at this temperature.

In view of spreading desertification (Huang et al., 2016) high interest exists in better estimating the role of atmospheric desert

aerosol for the ice phase in clouds and thus on the aerosol indirect effect. The objective of this study is to quantify INP concentrations in freshly emitted dust plumes close to the Sahara and the role of the composition of the desert aerosol on ice nucleation. This study was part of the "Cloud Affecting particLes In Mineral dust from the sAhara" (CALIMA) campaigns which took place at Izaña in late July and August of 2013 and 2014. In the following, we give an overview over the two campaigns and describe our methods to measure INPs and aerosol size distribution and composition. We report INP concentrations at different

temperature and relative humidity conditions. Furthermore, we investigate the effect of particle size and surface area on INP concentrations in different air masses as well as the role of fluorescent biological particles (FBAPs) and bulk chemical composition for ice nucleation. We discuss how representative our measurements are considering the technical limitations of our ice nucleation chamber PINC and compare our results to two common ice nucleation parameterization schemes from the literature.



## 2 Methods

### 2.1 Site description

The two CALIMA campaigns took place from 30 July to 29 August 2013 and from 23 July to 27 August 2014 at the Izaña Atmospheric Observatory ($16°29'58''$ W, $28°18'32''$ N), located at 2373 m above sea level (asl.) in Tenerife, Spain. The location usually remains above the stratocumulus layer typical for the subtropical oceanic boundary layer (Rodríguez et al., 2009) and is representative for the free troposphere during nighttime. During daytime, orographic upward flows transport water vapor and trace gases from the boundary layer to the location of the observatory (Rodríguez et al., 2009) which may result in new particle formation (García et al., 2014). During the summer, the observatory is frequently located within the SAL which carries large amounts of dust from North Africa over the Atlantic ocean (Rodríguez et al., 2015). Further details about the meteorological characteristics can be found in Rodríguez et al. (2009), Carrillo et al. (2015) and references therein.

### 2.2 Ice nucleating particle concentration measurements

During both CALIMA campaigns, ice nucleating particle concentrations ($[INP]$) were measured with the Portable Ice Nucleation Chamber (PINC, Chou et al. 2011; Kanji et al. 2013; Boose et al. 2016). PINC follows the physical principal of a Continuous Flow Diffusion Chamber (CFDC, Rogers 1988; Rogers et al. 2001). The aerosol sample is drawn through a chamber between two ice-coated walls at different subzero temperatures which provide supersaturated conditions with respect to (wrt.) ice. If the onset conditions of an INP are reached an ice crystals grows on the aerosol particle. Measurements were carried out at temperatures ($T$) ranging from 233-258 ($\pm$ 0.4) K and relative humidities wrt. ice ($RH_i$) between 100-150 ($\pm$ 2) %. Ice nucleation in the deposition regime, where ice forms directly from the vapor phase, was inferred by conducting experiments below water saturation. Above water saturation condensation freezing, where ice starts forming while water vapor condenses on an INP, as well as immersion freezing, where the INP is immersed in a droplet prior to initiating freezing, were investigated. The latter two processes cannot be distinguished with our method and are thus only referred to as condensation freezing. Measurements in the deposition ($RH_w$ = 92 %) and condensation regime ($RH_w$ = 105 %) were conducted most often at 241 K during the campaign. Furthermore, scans of $RH$ at different temperatures were performed, starting at $RH_i$ = 100 % up to $RH_w \geq 100$ %. INP concentrations at standard temperature and pressure (STP, $T$ = 273.15 K, $p$ = 1013 hPa) and PINC $T$, $RH_i$ and $RH_w$ data were averaged over 1 min intervals. INP concentrations are given in standard liters (stdl$^{-1}$). Before and after each experiment, the sample flow is drawn through a filter to measure the background INP concentration of the chamber which is subtracted from the measured INP concentation during analysis.

Due to the low number of INPs (down to below 1 in $10^6$ particles) in the atmosphere, their statistical counting uncertainties are determined based on Poisson statistics (Rogers et al., 2001; Boose et al., 2016). The limit of detection (LOD) equals the error of the background concentration. To lower the LOD of PINC, an aerodynamic lens concentrator (Enertechnix Inc.; Seattle, USA; Novosselov and Ariessohn 2014) was installed upstream of PINC. The concentration factor for INPs was determined as 4.3 $\pm$ 2 by routinely comparing $[INP]$ of periods when the concentrator was off to periods when it was on. An impactor with an aerodynamic $D50$ cut-off diameter of 0.9 $\mu$m (diameter at which 50 % of the particles impact) was used upstream



of PINC to allow a distinction by size of larger ice crystals which had formed in PINC and unactivated aerosol particles and droplets. Ice crystals, droplets and aerosol particles in the size range 0.5-25 $\mu$m were detected with an Optical Particle Counter (OPC; Lighthouse REMOTE 5104; Fremont, USA) downstream of PINC. Particles larger than 3 $\mu$m were classified as ice crystals. The effects of the impactor and the concentrator on the INP measurements and their representativeness for ambient

INP concentrations are discussed in the results section.

## 2.3   Aerosol particle measurements

### 2.3.1   Aerosol size distribution

Aerosol particle size distributions and concentrations are monitored continuously at Izaña within the framework of the Global

Atmosphere Watch program of the World Meteorological Organization (Rodríguez et al., 2015). Number concentrations of particles larger than 0.01 $\mu$m are determined with a Condensation Particle Counter (CPC; TSI; model 3776). Mobility particle diameter ($d_m$) between 0.01-0.44 $\mu$m is measured with an Scanning Mobility Particle Sizer (SMPS; TSI; DMA model 3081, CPC model 3010) and the aerodynamic diameter ($d_{aer}$) between 0.5-20 $\mu$m with an Aerodynamic Particle Sizer (APS; TSI; model 3321). The size distributions obtained by the SMPS and APS were merged and the mobility and aerodynamic diameter

were converted to volume equivalent diameter ($d_{ve}$). For that, the shape factor and particle density were determined on a daily basis from the known dust concentration and by optimizing the size distribution overlap. All concentrations are given at STP conditions, i.e. standard cm$^{-3}$ (std cm$^{-3}$) and in a specific size range $x$ ($N_x$). Aerosol size distribution data were not available during the first two days of August 2013.

### 2.3.2   Bulk chemical composition

Chemical characterization of Total Particulate Matter (PM$_T$) and particulate matter smaller than 10 $\mu$m (PM$_{10}$), 2.5 $\mu$m (PM$_{2.5}$), and 1 $\mu$m (PM$_1$) aerodynamic diameter, collectively referred to as PM$_x$ hereafter, was performed in samples collected during the two CALIMA campaigns. To avoid the daytime upward flows from the boundary layer (Rodríguez et al., 2009), these PM$_x$ samples were collected at night (22:00-06:00 UTC), when free tropospheric airflows prevail, as part of the longterm aerosol chemical composition program started in 1987 (Rodríguez et al., 2012). A total of 30 and 26 PM$_T$, 31 and 32

PM$_{10}$, 31 and 30 PM$_{2.5}$ and 31 and 30 PM$_1$ nocturnal samples and additionally 12 and 11 PM$_{2.5}$ daytime (10:00-16:00 UTC) samples were collected during CALIMA2013 and CALIMA2014, respectively.
Samples were collected on quartz microfiber filters ($d$ = 150 mm) using high volume (30 m$^3$h$^{-1}$) samplers. PM$_x$ concentrations were determined by conditioning the filters at 293 K and 30 % $RH$, applying the EN-14907 gravimetric procedure (except for $RH$ set to 30 % instead of 50 %). Chemical characterization included elemental analysis by inductively coupled plasma atomic

emission spectrometry and inductively coupled plasma mass spectrometry (e.g.: Al, Fe, Ca, K, Mg, Na, Ti, V, Ni), anions by ion chromatography (NO$_3^-$, SO$_4^{2-}$, and Cl$^-$, ammonium by selective electrode (NH$_4^+$) and organic (OC) and elemental (EC) carbon by the themo-optical transmittance method (see details on the program in Rodríguez et al. 2012; Rodríguez et al. 2015).





Chemical characterization was used for a mass closure of PM$_x$ (see Table 1). Nitrate occurred mostly in the supermicron fraction, whereas ammonium was found in the submicron range, indicating that the latter is associated with sulfate. Concentrations of sulfate versus ammonium in the submicron aerosol samples showed a high correlation and linearity (R$^2$ = 0.89). The fit line has a slope of 3.39, much closer to the theoretical ratio of sulfate to ammonium in ammonium sulfate (= 2.66) than in

ammonium bisulfate (= 5.33). Hence, we split the observed sulfate in two fractions: ammonium sulfate (a-SO$_4^{2-}$) and non-ammonium sulfate (na-SO$_4^{2-}$). NO$_3^-$ and na-SO$_4^{2-}$ were assumed to be present as Ca-salts and the remaining Ca to be present as carbonate. From earlier analysis of dust samples at Izaña we determined a ratio of Si/Al = 2 (Kandler et al., 2007) and that 40 % of the observed iron is present as oxide (Lázaro et al., 2008). The dust mass was then calculated as the sum of Al$_2$O$_3$ + Fe + SiO$_2$ + CaCO$_3$ + Fe$_2$O$_3$ + Ti + Sr + P + K + Na + Mg and then normalized such that Al accounts for 8 % of the dust,

i.e. the mean earth crust value. More details are provided in Rodríguez et al. (2012). The undetermined fraction of PM, i.e. the difference between the gravimetrically determined PM and the sum of the chemical compounds, was significantly higher in PM$_1$ and PM$_{2.5}$ than in PM$_{10}$ and PM$_T$ which has been observed in earlier studies (Ripoll et al., 2015). It is attributed to water residuals not fully removed during filter conditioning.

Hourly values of PM$_{2.5}$ and PM$_{10}$ concentrations were calculated by multiplying the aerosol volume concentrations, derived

from the APS size distributions, with experimentally determined volume-to-mass conversion factors (density equivalent) as described in Rodríguez et al. (2012). PM$_x$ values are given per standard m$^{-3}$ (std m$^{-3}$). Measurements of the absorption and scattering coefficients continuously performed at Izaña within the framework of GAW, were used to identify biomass-burning aerosol.

**2.4 Fluorescent biological aerosol particles**

During CALIMA2014, size resolved fluorescent biological aerosol particle (FBAP) concentration was measured with a Waveband Integrated Bioaerosol Sensor (WIBS-4; Kaye et al. 2005; Toprak and Schnaiter 2013). The WIBS-4 makes use of the UV light-induced fluorescence (UV-LIF) method where the auto-fluorescence in two spectral bands (320-400 nm and 410-650 nm) of the particles is measured after subsequent illumination with laser pulses at 280 and 370 nm, resulting in the three detection

channels F1 (excitation at 280 nm and detection in 320-400 nm), F2 (excitation at 280 nm and detection in 410-650 nm), and F3 (excitation at 370 nm and detection in 410-650 nm). These excitation and detection wavelengths were chosen such that typical components of biological particles (e.g. coenzymes such as NADH, proteins or amino acids such as tryptophan, Pöhlker et al. 2012) can be detected. In the present study, we used the simultaneous fluorescence in channels F1 and F3 of WIBS-4 as the criterion for the detection of FBAPs (Toprak and Schnaiter, 2013). However, also non-biological particles such as mineral dust

can exhibit simultaneous fluorescence in these two channels, resulting in a residual fraction of misclassified FBAP particles. Several mineral dust samples thus have been examined previously in the laboratory to find threshold values for each detection channel and their combinations to distinguish FBAP from mineral dust particles. With this method, a small percentage of particles can still be wrongly classified. The highest cross-sensitivity was found for a pure feldspar sample ($N_{FBAP}/N_{tot}$ = 1.5 %). Other mineral dust samples (Illite and Arizona Test Dust) showed a much lower cross-sensitivity ($N_{FBAP}/N_{tot} \leq$ 0.1 %).



### 2.5 Coupling of PINC - PCVI - WIBS

In order to study the fluorescence and thus biological content of INPs directly, PINC and WIBS were occasionally coupled during CALIMA2014. Downstream of the PINC OPC, a Pumped Counterflow Virtual Impactor (PCVI, model 8100, Brechtel Manufacturing Inc., USA; Boulter et al. 2006; Kulkarni et al. 2011) was installed to solely select ice crystals while omitting

the smaller, unactivated aerosol particles and droplets. The crystals then were warmed up to room temperature and dried and the remaining residuals were sampled by the WIBS. As the overlapping size of particles which pass the impactor upstream of PINC ($\leq 0.9$ $\mu$m) and which are measured with full efficiency by the WIBS ($\geq 0.8$ $\mu$m) was very restricted, for these periods the impactor upstream of PINC was replaced by a cyclone (URG-2000-30EG, URG Corporation, Chapel Hill, NC USA) with a cut-off diameter of 3.5 $\mu$m at a volumetric flow of 12 lpm. It was confirmed by tests at $RH_i$ = 100 % that no aerosol particles

entered which were in the size range of the ice crystals and could thus be miscounted as INPs. Before each experiment, the PCVI pump and add flow were adjusted such that for a period of about 5 minutes no particles were counted with a Condensation Particle Counter (TSI, model 3772) behind the PCVI at ice saturation but only at supersaturated conditions wrt. ice. Thus, the PCVI cut-off was set to a size above the largest aerosols and droplets and below the ice crystal size range. This yielded a pump volume flow of 13.4 lpm and an add flow of 2.8 lpm. A dilution flow of 1.2 lpm was added downstream of the PCVI to meet

the required 2.5 lpm WIBS flow. A description of the characterization of the PCVI can be found in the Supplementary Material and in Kupiszewski et al. (2015).

### 2.6 Back trajectories

10-day air mass back trajectories were calculated with the Lagrangian model LAGRANTO (Wernli and Davies, 1997). ECMWF reanalysis data were used as input and the model was run with a resolution of 0.25°. To best capture bifurcations, trajectory end

points were set to the location and altitude of the Izaña observatory as well as 0.5° north, south, west and east and $\pm$ 50 hPa, similar to the method described in Boose et al. (2016).

### 2.7 Data analysis

It has been shown that INPs can differ largely in size, depending on the environment (Mason et al., 2016). In dusty environments as in the present study, INPs are rather large (see Section 3.3) whereas in clean marine air, the majority of INPs might be

$0.02 \leq d \leq 0.2$ $\mu$m (Bigg and Miles, 1963; Wilson et al., 2015). Furthermore, heterogeneous ice nucleation is a surface area dependent process (Fletcher, 1958). The number of ice nucleation active sites per particle surface area, $n_s$, (DeMott, 1995; Connolly et al., 2009; Hoose and Möhler, 2012; Niemand et al., 2012) is a simplified concept to quantify the several proposed effects which lead to ice nucleation (Pruppacher and Klett, 1997). It is calculated as (Hoose and Möhler, 2012):

$$n_s(T, RH_i) = -\frac{\ln\left(1 - AF(T, RH_i)\right)}{\overline{A}_{ve}} \approx \frac{AF(T, RH_i)}{\overline{A}_{ve}} = \frac{INP(T, RH_i)}{A_{tot}} \qquad (1)$$



where $\overline{A}_{\mathrm{ve}}$ and $A_{\mathrm{tot}}$ are the average and total volume equivalent aerosol surface area, respectively, and $AF = [INP]/N_{\mathrm{tot}}$ the ratio of INP concentration to total aerosol particle concentration. The approximation is only valid for $AF \leq 0.1$ which was the case throughout the field study. We calculated $n_{\mathrm{s}}$ by integrating the surface area of each size bin, assuming $A_{\mathrm{ve}} = \pi d_{\mathrm{ve}}^2$, over the full size range of the volume equivalent diameter, $d_{\mathrm{ve}} = 0.02\text{-}20\ \mu\mathrm{m}$. For calculating $n_{\mathrm{s}}$ as well as $AF$ the particle losses due to the impactor and the particle enrichment due to the concentrator were accounted for based on laboratory characterization measurements.

# 3   Results and discussion

## 3.1   The CALIMA2013 and 2014 campaigns: an overview

The two CALIMA campaigns differ in frequency and amount of dust being present at the observatory. Figure 1 shows the Aerosol Optical Depth (AOD) over the North Atlantic, averaged for the time period of CALIMA2013 and CALIMA2014, respectively. Table 1 shows the mean chemical composition and mass closure of $PM_T$ during both campaigns. Mean $PM_T$ was 99 $\mu$g std m$^{-3}$ during CALIMA2013 and 52 $\mu$g std m$^{-3}$ in 2014, consistent with the satellite observations of the Saharan Air Layer. During CALIMA2013, the SAL was on average expanded northward over the Canary Islands (Fig. 1a), whereas during CALIMA2014 the SAL frequently occurred along a narrow corridor between 14° and 24°N, i.e. south of the Canary Islands (Fig. 1b). The dust load at Izaña is correlated with a northward (high load) - southward (low load) shift of the SAL associated with the variability of the North African dipole intensity, i.e. the intensity of the Saharan high compared to the monsoon tropical low (Rodríguez et al., 2015).

In the SAL, $PM_T$ is to over 90 % constituted by dust, which is mixed with low amounts of ammonium sulfate, nitrate and organic matter, each accounting for 0.1-1 % of $PM_T$ (Table 1a). This is also true for $PM_{10}$ (Table 1b), given that $PM_T$ is mostly consituted by $PM_{10}$. Under dust-free conditions, $PM_{10}$ is very low ($< 3$ $\mu$g std m$^{-3}$). Therefore, the hourly $PM_{10}$ records are a good proxy of hourly bulk dust$_{10}$, i.e. concentrations of dust particles smaller than 10 $\mu$m. In the smaller size ranges, mineral dust is also dominant, accounting for 70 % and 60 % of $PM_{2.5}$ and for 60 % and 30 % of $PM_1$ during CALIMA2013 and 2014, respectively. Following Adam et al. (2010), we classified Saharan dust events with $PM_{10} \geq 100$ $\mu$g std m$^{-3}$ as major (mSDE), with $50 \leq PM_{10} \leq 100$ $\mu$g std m$^{-3}$ as intermediate (iSDE) and with $10 \leq PM_{10} \leq 50$ $\mu$g std m$^{-3}$ as minor dust events.

Figure 2 and 3 show time series of INP and aerosol concentrations during CALIMA2013 and CALIMA2014, respectively. The first days of CALIMA2013 were subject to an extreme dust event with $PM_{10}$ values of 100-700 $\mu$g std m$^{-3}$ (1-3 Aug., Fig. 2c, mSDE1), followed by a second, smaller but still major dust event of $PM_{10} = 100\text{-}200$ $\mu$g std m$^{-3}$ (3-6 Aug., mSDE2). During the following weeks, Izaña was within the SAL most of the time, with $PM_{10}$ values of 50-100 $\mu$g std m$^{-3}$ (6-13 Aug., iSDE1 and iSDE2), 25-50 $\mu$g std m$^{-3}$ (13-19 Aug.) and 100-250 $\mu$g std m$^{-3}$ (19-25 Aug., mSDE3 and mSDE4). Dust free conditions due to North Atlantic air masses prevailed the last days of the campaign, with $PM_{10}$ value of 0.1-3 $\mu$g std m$^{-3}$ (25-30 Aug.). During this period, also a biomass burning event caused by wildfires in North America was detected (27-28 Aug., BB1). The first days of the CALIMA2014 campaign (Fig. 3) had low $PM_{10}$ values of 0.1-2 $\mu$g std m$^{-3}$ (24 Jul-5 Aug, Fig. 3c) during



north-westerly incoming flow from the Atlantic and North America, including a long-range transported biomass-burning event (25-30 Jul, BB1). An intermediate dust event (5-8 Aug, iSDE1) with $PM_{10} \leq 60$ $\mu$g std m$^{-3}$ was followed by prevailing dust free conditions (8-17 Aug, $PM_{10} = 0.1$-3 $\mu$g std m$^{-3}$). The end of the campaign experienced higher dust impact, with three iS-DEs (iSDE2: 17-19 Aug, 50-95 $\mu$g std m$^{-3}$; iSDE3: 21-22 Aug, 30-70 $\mu$g std m$^{-3}$ and iSDE4: 23-24 Aug, 30-75 $\mu$g std m$^{-3}$)

5 as well as two major dust events (mSDE1: 19-20 Aug., 100-280 $\mu$g std m$^{-3}$ and mSDE2: 26-27 Aug., 150-230 $\mu$g std m$^{-3}$). Dust free conditions prevailed from 25-26 Aug. ($PM_{10} = 0.1$-3 $\mu$g std m$^{-3}$).

Size distribution measurements showed that (i) the number of particles with $d_{ve} \geq 0.5$ $\mu$m (Fig. 2d and Fig. 3d) tracks dust events; (ii) during biomass burning events, the concentration of particles with $0.5 \leq d_{ve} \leq 1$ $\mu$m increased but did not lead to an elevation of $PM_{10}$ levels; (iii) the increase in the height of the planetary boundary layer and new particle formation during

10 day time is visible by the daily oscillation of the concentration of particles with $d_{ve} \leq 0.5$ $\mu$m (Fig. 3e) during periods of low to no dust (e.g. 9-17 August 2014). At night time, the observatory is located in the free troposphere and particle concentration decreases but shows (iv) higher values during biomass burning periods due to an increase in the free tropospheric background. And lastly (v), during dust events the concentration of particles $d_{ve} \leq 0.1$ $\mu$m is reduced and the daily variation vanishes (Fig 2e and Fig 3e) as the larger dust particles serve as coagulation sink for them (García et al., 2014).

### 3.2 Ice nucleating particle concentrations

The higher frequency and intensity of the dust events during CALIMA2013 in comparison to CALIMA2014 is reflected in the average INP concentrations. Mean condensation mode $[INP_{241K, 105\% RH_w}] \pm \sigma$ were $229 \pm 468$ stdl$^{-1}$ in 2013 and $23 \pm 43$ stdl$^{-1}$ in 2014 and mean deposition mode $[INP_{241K, 92\% RH_w}]$ were $1.5 \pm 2.3$ stdl$^{-1}$ in 2013 and $1.2 \pm 1.1$ stdl$^{-1}$ in

2014. The time series of $[INP]$ in the condensation mode at 241 K (Fig. 2a and 3a) show that generally INP concentrations increased during dust events. During the extreme dust event in 2013 (mSDE1), $[INP_{241K, 105\% RH_w}] \geq 2500$ stdl$^{-1}$ were observed. During the biomass burning events on the other hand, $[INP_{241K, 105\% RH_w}]$ stayed below 10 stdl$^{-1}$, which is comparable to those during clean background conditions when air masses came from over the North Atlantic.

Deposition mode $[INP]$ time series are shown in Fig. 2b and 3b. $[INP_{241K, 92\% RH_w}]$ were in general lower than those in con-

25 densation mode at the same temperature. The mSDEs led to an increase in $[INP_{241K, 92\% RH_w}]$ to up to 32 stdl$^{-1}$ but the iSDEs hardly influenced the $[INP]$.

As shown by Boose et al. (2016), measurements of ambient INP concentrations are significantly biased towards too high values if a large number of data points falls below the limit of detection (LOD) of the INP counter. There is no standardized method to account for these sub-LOD measurements. In Table 2 we therefore report the average $[INP]$ during CALIMA2014

at different $T$ and $RH$-conditions in three ways: 1) excluding all $[INP] \leq LOD$, 2) including $[INP] \leq$ LOD and setting $[INP_{\leq LOD}] = LOD$ and 3) including $[INP] \leq$ LOD and setting $[INP_{\leq LOD}] = 0$. The last column contains the maximum percentage of this theoretical positive bias of the reported INP concentrations due to this LOD-effect. This value is highest (up to 81 %) for warm $T$ and low $RH$ and becomes 0 at $T = 233$ K and $RH_w = 100$ %. For the temperature and $RH$ conditions that we focus on in the following, $[INP_{241K, 92\% RH_w}]$ and $[INP_{241K, 105\% RH_w}]$, the maximum bias due to excluding data below the





LOD is 78 % and 15 %, respectively. Currently, there is no commonly used method for these type of observations to account for sub-LOD data. Thus, to stay comparable to other observations the data below detection limit is excluded in the following analysis.

### 3.3 Ice nucleating particle dependency on size

Several earlier studies have shown that the efficiency of INPs of the same type to nucleate ice increases with the size of the INPs (Berezinski et al., 1988; Welti et al., 2009; Kanji and Abbatt, 2010). DeMott et al. (2010) showed that ambient $[INP]$ could be parameterized by using the concentration of aerosol particles with $d \geq 0.5\,\mu$m and temperature. This was further supported by Chou et al. (2011) who observed a better correlation of ambient $[INP]$ in deposition mode at 241 K with aerosol particles of $0.5 \leq d_{\mathrm{aer}} \leq 0.6\,\mu$m ($R^2 = 0.88$) than with particles of $0.3 \leq d_{\mathrm{aer}} \leq 0.5\,\mu$m ($R^2 = 0.69$). Mason et al. (2016)
found that a large fraction (40-95 % at 248 K) of INPs in ground-based measurements were larger than 1 $\mu$m in diameter.

We investigated the correlation of $[INP]$ and aerosol particles of different sizes during CALIMA2014. Fig. 4 shows the resulting $R^2$ values for different size bins and $[INP_{\mathrm{241K,\ 105\%\ RH_w}}]$ for all periods and dust periods only. Generally, the $R^2$ is higher if only the dust periods are taken into account. Already for particles of 0.1-0.2 $\mu$m the correlation is fairly good ($R^2 = 0.5$) for the dust periods. With increasing aerosol size, the dust aerosol dominates the aerosol load more and more and the $R^2$ values
converge. For the dust periods, the $R^2$ stays approximately constant at sizes $\geq 0.3\,\mu$m. The 0.1 $\mu$m threshold found for the dust dominated aerosol corresponds to the lower size limit found by Marcolli et al. (2007). This does not necessarily imply that the atmospheric INPs at T = 241 K are as small as 0.1 $\mu$m, however, it highlights that particles smaller than 0.5 $\mu$m also need to be considered relevant for atmospheric ice nucleation when dust is present.

Comparing the $R^2$ of $[INP_{\mathrm{241K,\ 105\%\ RH_w}}]$ with $N_{\mathrm{0.5\text{-}1\,\mu m}}$ versus that with $N_{\mathrm{0.5\text{-}20\,\mu m}}$ shows that the upper size limit of particles
entering PINC of 1 $\mu$m only has a minor effect on the correlations with $[INP]$ compared to all particles of $0.5 \leq d_{\mathrm{ve}} \leq 20\,\mu$m. Figure 4 also shows that $[INP_{\mathrm{241K,\ 92\%\ RH_w}}]$ correlates only very weakly ($R^2 = 0.14$) with aerosol particles of $0.5 \leq d_{\mathrm{ve}} \leq 1\,\mu$m and even less with particles of smaller sizes. This corresponds to the observation that only the mSDEs led to a noticeable increase of deposition mode $[INP_{\mathrm{241K,\ 92\%\ RH_w}}]$.

### 3.4 Ice nucleating particle dependency on surface area

By comparing $[INP]$, $AF$ and $n_{\mathrm{s}}$, number and size related effects on ice nucleation can be segregated. Fig. 5 shows scans of $RH_{\mathrm{i}}$ at three different constant temperatures $\leq 248$ K at different times during CALIMA2014. At 253 K and $RH_{\mathrm{i}} \leq 130$ % no $[INP]$ above the detection limit was observed (not shown). The scans during mSDE1 led to more than a factor of 8 times higher $[INP]$ at 233 K, 240 K and 248 K (Fig. 5a, b and c) compared to the non-dust background periods (BG), the biomass burning
period, as well as the other dust events. This is in part simply due to the high number of particles as seen in the $AF$ shown in Fig. 5d - f. The differences between the mSDE1 and scans during other periods get smaller compared to the differences in $[INP]$. At 240 K the scan during mSDE2 shows a comparable high $AF$ as that during mSDE1. At last, the $n_{\mathrm{s}}$ in Fig. 5g - i reveals that during dust dominated periods the aerosol particles are more ice-active even when the higher concentration and larger





surface area are accounted for. In addition, differences between the different SDEs can be found by up to an order of magnitude which must be related to the composition of the aerosol particles. These factors will be discussed in the following sections.

### 3.5 Biological aerosol particles as INP

In this and the following section we investigate the dependence of $[INP]$ on the biological content of single aerosol particles and the bulk chemical composition.

Indication of an enrichment of FBAP during Saharan dust events compared to non-dust periods was determined by WIBS measurements at the Jungfraujoch in the Swiss Alps (Kupiszewski et al., 2015). During CALIMA2014, WIBS measurements were conducted at the Izaña observatory to study how much FBAP the desert aerosol contains already close to its emission source and what effect this has on ice nucleation. Figure 6 shows the time series of $[INP_{241K, 105\% RH_w}]$ and $[INP_{241K, 92\% RH_w}]$ during CALIMA2014 together with that of fluorescent particles ($N_{FBAP}$) and total aerosol particles ($N_{tot}$) of $0.8 \leq d_p \leq 20\,\mu m$ as measured by the WIBS as well as the ratio of the latter two.

The black and green data points in Fig. 6c are the ambient $N_{tot}$ and $N_{FBAP}$, respectively, measured in parallel to PINC. It can be seen, that during dust events, both, ambient $N_{tot}$ as well as ambient $N_{FBAP}$, increased, i.e. there were more FBAPs during SDEs than during non-dust times. Yet, the ratio of $N_{FBAP}/N_{tot}$ (black data points in Figure 6d) decreased, showing that the fraction of FBAPs is lower in the desert aerosol than it is for the non-dust dominated aerosol.

The fluorescent and total INP concentrations measured by the WIBS downstream of PINC, $N_{FBAP, INP}$ and $N_{tot, INP}$ (magenta and purple data points in Fig. 6c), were much lower than those of the ambient aerosol because only few particles act as INP. The higher ratio of $N_{FBAP,INP}/ N_{tot,INP}$ (purple data points in Fig. 6d) compared to the ambient $N_{FBAP}/N_{tot}$ right before or after the PINC-PCVI-WIBS coupled measurements, shows that more fluorescent particles were found in the INPs compared to the ambient aerosol. Up to 25 % of the INPs measured with the WIBS were FBAPs, also during SDEs. In contrast, a maximum fraction of 20 % of the ambient aerosol particles were fluorescent during non-dust periods and $\leq 5$ % during dust events.

It should be kept in mind that the counting statistics for the WIBS measurements downstream of PINC were low due to the generally low number of INPs and the restriction of the PINC-PCVI-WIBS coupling to only three measurement intervals of a few hours each during CALIMA2014. To study the relationship of FBAPs and $[INP]$ in more detail, we correlated $[INP]$ to ambient $N_{tot}$ and $N_{FBAP}$ measured by WIBS in parallel. Figure 7a depicts a very good correlation of $N_{tot}$ with $[INP_{241K, 105\% RH_w}]$ ($R^2 = 0.91$) and Fig. 7c a fairly good correlation of $N_{FBAP}$ with $[INP_{241K, 105\% RH_w}]$ ($R^2 = 0.49$). The correlations of deposition mode $[INP_{241K, 92\% RH_w}]$ with both $N_{tot}$ and $N_{FBAP}$ are much weaker ($R^2 = 0.31$ and $0.18$, see Fig. 7b and d). Figure 7c furthermore shows that there were not enough FBAPs to explain all observed condensation mode $[INP_{241K, 105\% RH_w}]$, as $N_{FBAP} \leq 70$ stdl$^{-1}$, hence about a factor 4 less $N_{FBAP}$ than INPs. This agrees well with the maximum ratio of 25 % of $N_{FBAP}/N_{tot}$ found for INPs (Fig. 6d) and is likely also why condensation $[INP]$ at 241 K weakly anticorrelate with the $N_{FBAP}/N_{tot}$ ratio of the ambient aerosol ($R^2 = 0.12$) in Fig. 6e. Even though FBAPs are enriched in the INPs compared to the ambient aerosol, their concentration is too low to be the dominant INP type. For the deposition mode $[INP]$, the $N_{FBAP}$



concentration would be sufficient but the correlations are so weak that a predominant role of FBAP as INP is unlikely.

Part of the effectiveness of FBAPs to nucleate ice can be due to their often large size. Furthermore, the desert aerosol FBAPs can also be mineral dust particles with enough biological material on the surface to fluoresce such that these particles are classified as FBAPs. To exclude the size effect, we did the same analysis as above for $n_s$ instead of $[INP]$. The resulting correlation

coefficients for different conditions are given in Table 3. The correlation of $n_s$ with $N_{tot}$ and $N_{FBAP}$ is weaker than that for $[INP]$ showing that a large portion of the observed $[INP]$ can be explained by the size of the aerosol particles. However it also shows, that about 16 % ($R^2_{FBAP,ns} = 0.4^2$) of the variation of condensation $n_s$ is related to the concentration of FBAPs.

### 3.6 Aerosol chemistry and ice nucleation

The analysis of the relationship between $[INP]$ and the bulk chemical composition was done for nighttime measurements

only, when the aerosol chemistry was determined under the prevailing free tropospheric air masses. Figure 8 shows time series of nighttime averages of $[INP_{241K,\,105\%\,RH_w}]$, $[INP_{241K,\,92\%\,RH_w}]$, dust$_1$ and dust$_{10}$ during CALIMA2013 (Fig. 8a) and CALIMA2014 (Fig. 8b). In general, the averaged $[INP]$ in the condensation and deposition mode follow the dust$_1$ and dust$_{10}$ concentration. The scatter plots of $[INP]$ versus dust$_1$ presented in Fig. 9a and b, depict the fairly good positive correlation ($R^2 = 0.44$ for $[INP_{241K,\,105\%\,RH_w}]$ and $R^2 = 0.32$ for $[INP_{241K,\,92\%\,RH_w}]$ with dust$_1$). The samples collected within the Saharan

Air Layer (dust$_{10} \geq 10$ $\mu$g std m$^{-3}$, red) were segregated from those collected under dust free Atlantic air mass conditions (dust$_{10} < 10$ $\mu$g std m$^{-3}$, blue) for further analysis. The $[INP_{241K,\,105\%\,RH_w}]$ fall within a regime confined by the two dashed lines, $s_{min} = 2.95 \times 10^3$ INP $\mu$g$^{-1}$ and $s_{max} = 24.5 \times 10^3$ INP $\mu$g$^{-1}$ which represent the minimum and maximum concentration of INP per microgram of dust$_1$. We investigate how the chemical composition of the dust itself and the mixing of dust with pollutants influences the ratio INP/dust$_1$ between those limits. We observe more variability in the INP/dust$_1$ ratio for the

$[INP_{241K,\,105\%\,RH_w}]$ than for the $[INP_{241K,\,92\%\,RH_w}]$, similar to our findings for the size dependency and FBAPs.

$[INP_{241K,\,105\%\,RH_w}]$ showed a higher correlation with Al, Fe, Mg and Mn (R: 0.43-0.67) than with other elements (R: -0.1 to +0.4 for Ca, Na, and $CO_3^{2-}$). This is consistent with the idea that feldspar (Atkinson et al., 2013) and some clays (e.g. kaolinite, Yakobi-Hancock et al. 2013) may play a more relevant role as atmospheric INP than other minerals. The variability in dust composition is illustrated in Fig. 8c and d, which show the ratios of K, Mg, Ca and na-$SO_4^{2-}$ to Al in the dust samples collected

within the SAL. It can be seen that during the 15 days in 2013 when Izaña was permanently experiencing dusty conditions (1-15 Aug. 2013, Fig. 8c), the ratios varied significantly. This indicates different degrees of mixing between Mg-, K-, Ca- and na-$SO_4^{2-}$-containing minerals. For example, a high Ca and na-$SO_4^{2-}$ to Al ratio indicates the presence of evaporite minerals (e.g. calcite, gypsum or anhydrite) stemming from dry lake beds (Rodríguez et al., 2011). Although certain K-feldspars are considered to be more efficient INPs than clays (Atkinson et al., 2013), we did not find correlations between $[INP_{241K,\,105\%\,RH_w}]$

and a certain dust elemental composition (i.e. ratios to Al). This is likely due to the similar elemental composition of feldspars and clay minerals which are dominated by Al and Si and which makes it impossible to identify changes in their degree of mixing with the method used here.

Figure 8e and f show concentrations of nitrate and ammonium sulfate (a-$SO_4^{2-}$) in PM$_1$ and PM$_{10}$ together with $[INP]$. The concentrations of these pollutants showed a large variability during the dusty periods. Figure 9c shows the ratio $[INP_{241K,\,105\%\,RH_w}]$/dust$_1$





versus a-$SO_4^{2-}$ to Al, with Al as tracer of clays and feldspars. 10 out of 14 submicron dust samples (i.e. 71 %) collected in the SAL follow a linear trend ($R^2$ = 0.44). These samples are highlighted by open (2013) and filled (2014) red circles. No trend is found for deposition mode [$INP_{241K, 92\% RH_w}$]/dust$_1$ versus a-$SO_4^{2-}$/ Al (see Fig. 9d).

A possible explanation for this behavior is the weaker interaction with water molecules of large singly charged ions, such as
$NO_3^-$ and $NH_4^+$ compared to that of small ions with a high ionic charge density, such as $Al^{3+}$, $Mg^{2+}$, $Na^+$ or $Ca^{2+}$, often referred to as kosmotropes (Zangi, 2010). The low charge density ions (often referred to as chaotropes) are weakly hydrated, meaning that they bind weaker with water molecules than the hydrogen bonds of the water itself. This leads to an increase in entropy of the water near the ion and makes the water more mobile compared to pure water (Collins, 1997) and even more so compared to water close to a kosmotropic ion. We suggest this increase in mobility due to $NH_4^+$ ions at the surface of dust
particles allows the water molecules to rearrange easier compared to water molecules close to a pure dust surface. Thus, they can form an ice-like structure more easily as temperature decreases. This was similarly suggested for $K^+$ (weak chaotrope according to the definition by Collins 1997) versus $Ca^{2+}$ and $Na^+$ (kosmotropes) by Zolles et al. (2015) as an explanation of the higher (warmer) freezing temperature of K-feldspar particles compared to Ca- and Na-feldspars. $NH_4^+$ ions are more weakly hydrated than $K^+$, therefore we expect this effect to also be the case for ammonium sulfate on K-feldspar particles.
We infer that the a-$SO_4^{2-}$ exists as coating on the dust based on the observations by Kandler et al. (2007) who observed dust coated by sulfate for the submicron aerosol particles, a size range where we observe that sulfate is predominantly available as ammonium sulfate (accounting for 74 % of the total submicron sulfate). In the case of $RH_w$ = 92 % the particles are not diluted enough for freezing point depression to be negligible. Thus, an increase in the a-$SO_4^{2-}$/ Al ratio has a weak negative effect on [$INP_{241K, 92\% RH_w}$]/dust$_1$ as observed in our field measurements (Fig 9d).

Another possible explanation for the increase of [$INP_{241K, 105\% RH_w}$]/dust$_1$ with a-$SO_4^{2-}$/ Al is that a-$SO_4^{2-}$ suggests that the aerosol is more neutral, hence the acidity of the particles is reduced. As described earlier, several laboratory studies have observed a decrease in ice nucleation ability due to condensation of sulfuric acid which alters the dust surface.

Four outliers to the observed linear trend of [$INP_{241K, 105\% RH_w}$]/dust$_1$ versus a-$SO_4^{2-}$/ Al were indentified. The only point with a distinctly higher [$INP_{241K, 105\% RH_w}$]/dust$_1$ than the fit line (red filled triangle) occurred during the night 19-20 August
2014, when 15 $\mu$g std m$^{-3}$ of dust$_1$ and the highest average nighttime [$INP_{241K, 105\% RH_w}$] (367 stdl$^{-1}$) of the two CAL-IMA campaigns were recorded. We attribute this event to a higher fragmentation of the dust agglomerates (Perlwitz et al., 2015), i.e. a higher dust number to mass ratio than in other events of similar dust load. On 19-20 August 2014, the mean $N_{0.5-1 \mu m}$ = 11.6 std cm$^{-3}$ was 1.5 to 2 times that of events with similar dust$_1$ concentration (23 and 25 Aug. 2013 and 27 Aug. 2014: 15-17 $\mu$g std m$^{-3}$, $N_{0.5-1 \mu m}$ = 5.8-7.7  std cm$^{-1}$). Hence, the particle concentration $N_{0.5-1 \mu m}$ to dust$_1$ ratio was about
1.5 to 2 times higher than during similar high dust$_1$ event when [$INP_{241K, 105\% RH_w}$] ranged from 55-120 stdl$^{-1}$ (see Fig. 2a, c, d and 3a, c, d). The total surface area was also significantly larger on 19 August 2014 (1.8-2.7 $\times$ 10$^{-10}$ m$^2$ std cm$^{-3}$) compared to similarly high dust$_1$ days (0.8-2 $\times$ 10$^{-10}$ m$^2$ std cm$^{-3}$). As shown in Fig. 5h, the surface area alone could not fully explain the differences in observed [$INP$]. This indicates that the degree of fragmentation of the dust agglomerates (Perlwitz et al., 2015) influences the variability of the number of INP. The scattering of the $N_{0.5-1 \mu m}$ to dust$_1$ plot (Fig. 9f) illustrates the
variability in the dust agglomerates fragmentation.





The three outliers (red open triangles) that fall below the general trend in Fig. 9c (11, 18 and 24 of August 2013) are marked by rather low $dust_1$ (3, 5 and 9 $\mu g$ std m$^{-3}$) and $N_{0.5\text{-}1\,\mu m}$ (2.4, 2.7 and 7.0 std cm$^{-3}$). However, these are the only three dust events in both CALIMA campaigns, when the a-SO$_4^{2-}$/Al ratio was $> 1$. This suggests that either a significant fraction of the a-SO$_4^{2-}$ is externally mixed with the dust and consequently has a minor influence on the dust ice nucleation properties, or that

the higher ratio of a-SO$_4^v$/Al exceeds a threshold above which the a-SO$_4^{2-}$ reduces the ice nucleation ability of the dust particles potentially due to a depression of the freezing point in highly concentrated dust coatings. This is further supported by the deposition mode data in Fig. 9d which shows a weak decrease of $[INP_{241K,\,92\%\,RH_w}]/dust_1$ for the sample of a-SO$_4^{2-}$/Al $\geq 1$. In summary, the number of INP in the condensation mode at 241 K per $\mu g$ $dust_1$ varies within a factor of 7, i.e. from $2.95\times10^3$ to $24.5\times10^3$ $[INP_{241K,\,105\%\,RH_w}]$ $\mu g^{-1}$. The linear relationship between $[INP_{241K,\,105\%\,RH_w}]/dust_1$ and a-SO$_4^{2-}$/Al, which we

found in 71 % of the nighttime samples, suggests that mineral dust particles present in the $dust_1$ composition may experienced dust processing which led to enhanced mobility of the water molecules close to the dust surface by chaotropic ions and increased the ice nucleation ability of the dust. This relationship was not observed in the deposition mode ($[INP_{241K,\,92\%\,RH_w}]$).

### 3.7   Potentially omitted ice nucleating particles

A limitation in PINC arises from using an impactor to allow size-based differentiation of unactivated aerosol and INPs. Figure 10 shows the aerosol size distributions as measured for the ambient air as well as calculated for the expected size distribution sampled by PINC, i.e. after the concentrator and impactor, for different aerosol types dominating the air masses during CALIMA2014. Especially during the dust events, a significant fraction of the large particles are not sampled by PINC. As the ice nucleation ability usually increases with particle size, it is expected that a substantial fraction of INPs is missed and the ambi-

ent INP concentrations would be underestimated by this method when many large particles are present. To investigate this for our measurements at 241 K and 105 % $RH_w$, we calculated based on the measured size distributions and the characterization curves of the concentrator and the impactor, how many particles and accordingly which surface area were missed. With the $n_s$ value calculated based on the size distribution in the PINC chamber, and assuming $n_s$ to stay constant for larger particles, we determined the expected INP concentration which was omitted by our measurement method. A size independent $n_s$ has been

found for NX illite by Hiranuma et al. (2015).

A time series of the measured to total INP ratio during CALIMA2013 and 2014 is presented in Fig. 11. The ratio between measured and total assumed INPs varies a lot between 8-99 %. During dust events the measured INP ratio is generally lower, between 10-65 %, whereas during the CALIMA2014 biomass burning event basically all INP were captured. These findings are in line with a recent study by Mason et al. (2016) who found that about 40 % of INPs at 248 K were larger than 1 $\mu m$ at a

location at 2182 m altitude. Of course, our assumptions have several uncertainties. Apart from the measurement uncertainties, the assumption that particles above the PINC cut-off have the same $n_s$ as smaller particles might only be true if a significant composition dependence with size is absent in our samples, which we cannot confirm.





### 3.8 Predictability of INP concentrations close to the Sahara

Ice nucleation is still not understood well enough to be implemented in global climate models based entirely on theory. As a simplification, parameterizations are used, either based on Classical Nucleation Theory (Hoose et al., 2010; Ickes et al., 2016), laboratory experiments (Phillips et al., 2008; Niemand et al., 2012) or ambient observations (DeMott et al., 2010; Tobo et al.,

2013; DeMott et al., 2015). Since the current study is the first of its kind so close to the Sahara, we tested how well two of the ambient observation based parameterizations, namely DeMott et al. (2010) and DeMott et al. (2015), called D10 and D15 in the following, predict our observations. Both parameterizations are based on the ice nucleation temperature and the concentration of aerosol particles larger than $d = 0.5~\mu$m, $N_{>0.5}$, and predict $[INP]$ at 105 % $RH_w$. D10 was developed based on data from several ground-based and airborne studies in North America, Brazil and over the Pacific and includes aerosol of different type.

It takes the form:

$$[INP_T] = a(273.16 - T)^b (N_{>0.5})^{(c(273.16-T)+d)}, \tag{2}$$

with $a = 0.0000594$, $b = 3.33$, $c = 0.0264$ and $d = 0.0033$ (DeMott et al., 2010).

D15 follows the form of the parameterization by Tobo et al. (2013) but was particularly adapted for dust INPs based on data from two flights through dust-laden air layers over the Pacific and the U.S. as well as from laboratory data on dust samples.

The laboratory samples as well as the dusty air layers stemmed from the Sahara and Asia. D15 is given as:

$$[INP_T] = (cf)(N_{>0.5})^{(\alpha(273.16-T)+\beta)} \exp(\gamma(273.16 - T) + \delta), \tag{3}$$

with $\alpha = 0$, $\beta = 1.25$, $\gamma = 0.46$ and $\delta = -11.6$. $cf$ is a instrumental calibration factor specifically derived for the CFDC (DeMott et al., 2015). It is set to $cf = 1$ for the present study.

Figure 12 shows the predicted vs. the observed $[INP]$ at $RH_w$ = 105 % for both CALIMA campaigns, color-coded for dust

and biomass burning dominated periods. As input for the parameterizations $N_{0.5 \leq d_{ve} \leq 20~\mu m}$ were used. In Fig. 12a and c, INP concentrations as they were measured with PINC are compared to INP concentrations which were calculated using the ambient particle size distribution corrected for the effect of impactor losses and concentrator gains. Figure 12b and d show ambient concentrations. Here, the observed $[INP]$ include the omitted INPs, as described in the previous section. Thus, for the predicted $[INP]$ the ambient size distribution of particles between $0.5 \leq d_{ve} \leq 20~\mu$m was used without further corrections.

The blue data points are condensation mode $[INP]$ from an earlier campaign (CLACE2014) on the Jungfraujoch in the Swiss Alps described in Boose et al. (2016). During this campaign, measurements were conducted with PINC at $T$ = 241 K and $RH_w$ = 103 % in the winter time free troposphere.

For the CALIMA campaigns, D10 (Fig. 12a) has a median ratio of $[INP_{pred}]/[INP_{meas}]$ of 0.98 and predicts 50 % of the observed $[INP]$ within a factor of 5, and 60 % within an order of magnitude. Only during the biomass burning events, $[INP]$

are clearly overpredicted by about two orders of magnitude. D15 (Fig. 12c) generally overpredicts $[INP]$ by a median factor of 17. Only 5 % and 15 % of the $[INP]$ predicted by D15 fall within a factor of 5 and 10, respectively, of the observed $[INP]$. D15 works best for the major dust events and worst for the biomass burning events.

The Jungfraujoch data, shown in blue, are better predicted by both parameterizations. 60 % of the predicted $[INP]$ based





on D10 fall within a factor of 5 of the observed $[INP]$. For D15 this ratio lies at 81 % and 50 % even fall within a factor of 2. Both parameterizations agree significantly better with the observations at the Jungfraujoch than at Izaña. This, together with the fact that the field data in D10 and D15 were measured far away from the Sahara, suggests that the ice nucleation properties of the dust change between a location close to the Sahara and one with a much longer atmospheric transport time.

Especially comparing our results to D15, which was derived particularly for dust INP, suggests that dust particles measured close to the Sahara are less efficient than those which have been transported longer and experienced more atmospheric and cloud processing, such as dust arriving at the Jungfraujoch. However, the free tropospheric Jungfraujoch $[INP]$ were mostly below 10 stdl$^{-1}$, thus a comparison at higher aerosol particle and INP concentration is not possible. Similar measurements during Saharan dust events with a high dust load at the Jungfraujoch would therefore yield valuable insight into the role of

atmospheric aging on ice nucleation.

As we have shown in the last section, the ratio of INPs which are omitted by our measurement technique can vary greatly. We thus did the same comparison for our data including the omitted INPs. In the case of D10 (Fig. 12b) this hardly has an effect on the parameterization statistics: 48 % of the measured $[INP]$ from CALIMA are predicted within a factor of 5 and the median ratio of predicted/observed $[INP]$ is 0.7. The biomass burning event is again strongly overpredicted. For the Jungfraujoch

data, 77 % are predicted within a factor of 5. In Fig. 12d the median overprediction by D15 for the CALIMA data is a factor of 10, and 13 % of the predicted $[INP]$ are less than a factor of 5 different from the observed $[INP]$. For the Jungfraujoch data, the median ratio of predicted/observed $[INP]$ is 1.1. Hence, we conclude that the method of accounting for potentially omitted INPs does not significantly alter the results from the comparison to the D10 and D15 parameterizations.

## 4  Conclusions

For the first time, we presented INP concentrations in the Sahara Air Layer close to the Sahara from a total of 409 h, equivalent to 24.5 std m$^3$ of sampled air, of ground-based, on-line measurements conducted in July and August of 2013 and 2014 at 2373 m asl. in Tenerife, Spain. INP concentrations at temperatures between 233-253 K and relative humidities from $RH_i$ =100 % to $RH_w$ > 100 % were reported. They range from 0.2 stdl$^{-1}$ at $RH_w$ = 88 % up to 2500 stdl$^{-1}$ at 241 K and

$RH_w$ = 105 % during an extreme dust event in early August 2013. It was found that dust particles were the most efficient INPs at $T \leq 240$ K even if their higher number concentrations and larger surface area in comparison to the background aerosol was accounted for. At 248 K, the dust led to higher INP concentrations but similar $n_s$ compared to other aerosol types. At 253 K no significant ice nucleation was found at the investigated $RH_w$.

Submicron INP concentrations in the condensation mode at 241 K were observed to correlate well with the bulk dust mass

of particles smaller than 1 $\mu$m (dust$_1$) and furthermore with typical clay and feldspar tracers, such as Al, Fe, Mg and Mn (R: 0.43-0.67). They ranged between $2.95 \times 10^3$ and $24.5 \times 10^3$ INP per $\mu$g$^{-1}$ of dust$_1$. We studied how the mixing with pollutants affects the ice nucleation properties within these limits and if fluorescent biological material plays a role for the dust ice nucleation properties. The residuals of ice crystals from PINC were investigated by coupling the PINC with the use of a PCVI to




a WIBS which measures the single particle fluorescence and thus detects FBAPs. An indications was found for an enrichment of FBAPs in INPs compared to the total ambient aerosol. At maximum about 25 % of the observed INPs contained fluorescent biological material whereas less than 20 % of the total ambient aerosol particles were FBAPs and less than 5 % during Saharan dust event.

Furthermore, we observed that an increase in the ammonium sulfate to aluminum ratio correlates with the INP/dust$_1$ ratio. We suggest that the ammonium ions infer less with water molecules during the ordering process required for freezing compared to ions from mineral dust which have stronger interactions with water molecules thus inhibiting the ordering of water molecules to ice for the same temperature.

Lastly, we tested two common parameterizations from DeMott et al. (2010) and DeMott et al. (2015) which are based on field
measurements far away from the Sahara as well as laboratory dust measurements, on our data. The predicted $[INP]$ using D10 and D15 are significantly higher than the measured ones, especially for the parameterization from DeMott et al. (2015). The same comparison with data measured at the Jungfraujoch in the Swiss Alps showed that the ratio of predicted/ observed INP is much closer to 1. This could be an indication that atmospheric processing as it occurs during transatlantic or transeuropean advection of dust may enhance the ice nucleation ability of mineral dust compared to that after a relatively short atmospheric
transport between the Sahara and Tenerife. In this regard, further measurements of INP concentration and aerosol size distribution at the Jungfraujoch, especially during Saharan dust events, would further clarify the role of atmospheric aging on the efficiency of dust INP.

*Acknowledgements.* Y. Boose is funded by the Swiss National Science Foundation (grant 200020 150169/1). The research leading to these
results has received funding from the European Union's Seventh Framework Programme (FP7/2007-797 2013) under grant agreement n° 603445 (BACCHUS). Measurements at Izaña are part of the Global Atmosphere Watch program (funded by AEMET). This study was developed within the frame of the project AEROATLAN (CGL2015-66299) funded by the Minister of Economy and Competitiveness of Spain. M. I. García acknowledges the grant of the Canarian Agency for Research, Innovation and Information Society (ACIISI) co-funded by the European Social Funds. Analyses and visualizations of satellite data were produced with the Giovanni online data system, developed
and maintained by the NASA GES DISC. We also acknowledge the MODIS mission scientists and associated NASA personnel for the production of the data used in this research effort. The authors would like to express their gratitude to the technical team at Izaña and H. Wydler at ETH Zurich for their continuous support.



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





**Table 1.** Mass closure of $PM_x$ composition at Izaña during CALIMA 2013 and 2014. a) $PM_x$ concentrations determined by gravimetry. b) Major PM components, including sulfate as ammonium salt (a-$SO_4^{2-}$), nitrate ($NO_3^-$), ammonium ($NH_4^+$), organic matter (OM) and elemental carbon (EC). c) Selected dust components: non-ammonium sulfate (na-$SO_4^{2-}$), aluminum (Al), potassium (K) and iron (Fe).

| | | $PM_T$ | | $PM_{10}$ | | $PM_{2.5}$ | | $PM_1$ | |
|---|---|---|---|---|---|---|---|---|---|
| | | 2013 | 2014 | 2013 | 2014 | 2013 | 2014 | 2013 | 2014 |
| | | $\mu$g std m$^{-3}$ | | $\mu$g std m$^{-3}$ | | $\mu$g std m$^{-3}$ | | $\mu$g std m$^{-3}$ | |
| a) | $PM_x$ | 98.8 | 51.8 | 88.5 | 42.0 | 41.8 | 25.3 | 19.5 | 13.2 |
| b) | $dust_x$ | 95.8 | 48.4 | 82.0 | 39.2 | 29.9 | 14.7 | 11.3 | 3.7 |
| | a-$SO_4^{2-}$ | 0.9 | 0.4 | 0.9 | 0.3 | 0.8 | 0.3 | 0.9 | 0.3 |
| | $NO_3^-$ | 1.2 | 0.8 | 1.2 | 0.8 | 0.2 | 0.4 | 0.1 | 0.1 |
| | $NH_4^+$ | 0.3 | 0.1 | 0.3 | 0.1 | 0.3 | 0.1 | 0.3 | 0.1 |
| | OM | 1.2 | 1.5 | 1.2 | 2.1 | 1.2 | 3.7 | 1.2 | 1.5 |
| | EC | < 0.1 | < 0.1 | < 0.1 | < 0.1 | < 0.1 | < 0.1 | < 0.1 | < 0.1 |
| c) | na-$SO_4^{2-}$ | 2.5 | 0.7 | 1.9 | 0.7 | 0.9 | 0.3 | 0.3 | 0.3 |
| | Al | 7.6 | 3.9 | 6.8 | 3.2 | 2.5 | 1.2 | 0.9 | 0.3 |
| | K | 1.7 | 0.8 | 1.2 | 0.7 | 0.6 | 0.3 | 0.2 | 0.1 |
| | Fe | 4.0 | 2.0 | 3.3 | 1.7 | 1.2 | 0.6 | 0.4 | 0.1 |

**Table 2.** Average $[INP]$ during CALIMA2014, excluding data points below the limit of detection, including them and setting them to the LOD and including them and setting them to zero. The last column gives the maximum bias between columns 3 and 5.

| $T$ (K) | $RH_w$ (%) | $[INP]$ (stdl$^{-1}$) $> LOD$ | $[INP]$ (stdl$^{-1}$) $[INP_{\leq LOD}] \overset{!}{=} LOD$ | $[INP]$ (stdl$^{-1}$) $[INP_{\leq LOD}] \overset{!}{=} 0$ | max. bias |
|---|---|---|---|---|---|
| 233 | 92 | 39.9 | 26.6 | 26.7 | 0.3 |
| 233 | 100 | 192.6 | 192.6 | 192.6 | 0 |
| 238 | 92 | 3.35 | 1.74 | 1.53 | 0.54 |
| 238 | 102 | 26.5 | 24.7 | 24.7 | 0.07 |
| 240 | 105 | 22.6 | 19.3 | 19.2 | 0.15 |
| 240 | 92 | 1.52 | 0.61 | 0.33 | 0.78 |
| 242 | 92 | 1.39 | 0.66 | 0.46 | 0.67 |
| 242 | 102 | 26.5 | 22.5 | 22.5 | 0.15 |
| 248 | 80 | 0.80 | 0.40 | 0.15 | 0.81 |





**Table 3.** Correlation of $[INP]$ and $n_s$ with $N_{tot}$ and $N_{FBAP}$ as measured by the WIBS.

|  | R ($[INP_{241K, 105\% RH_w}]$) | R ($[INP_{241K, 92\% RH_w}]$) |
|---|---|---|
| $N_{tot}$ | 0.95 | 0.56 |
| $N_{FBAP}$ | 0.7 | 0.42 |
| $N_{FBAP}/N_{tot}$ | -0.35 | -0.23 |
|  | R ($n_{s,241K, 105\% RH_w}$) | R ($n_{s,241K, 92\% RH_w}$) |
| $N_{tot}$ | 0.65 | -0.14 |
| $N_{FBAP}$ | 0.40 | -0.27 |
| $N_{FBAP}/N_{tot}$ | -0.51 | -0.49 |

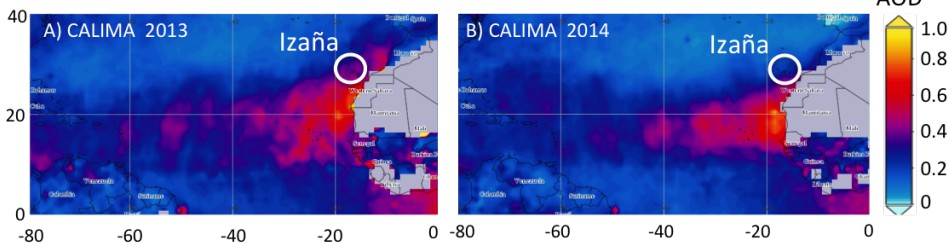

**Figure 1.** Mean MODIS Aerosol Optical Depth during CALIMA 2013 and CALIMA 2014.



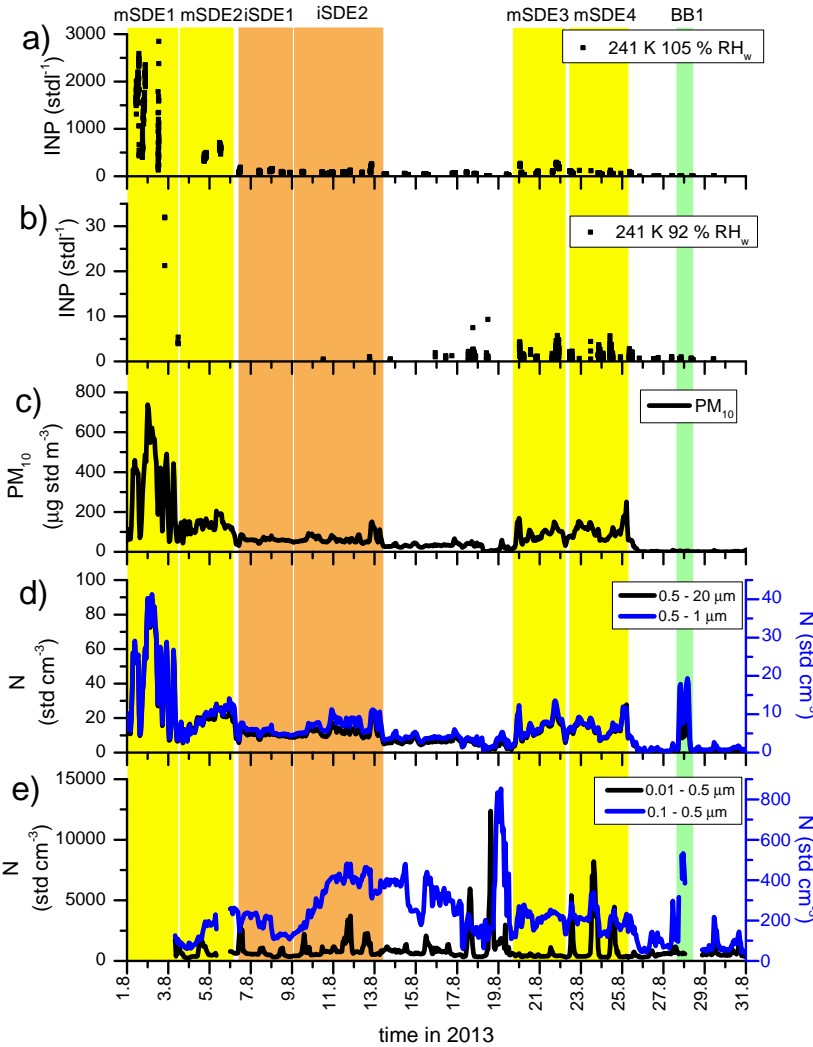

**Figure 2.** CALIMA2013: $[INP]$ in a) condensation and b) deposition mode at 241 K. c) $PM_{10}$, d) aerosol particle number concentration as measured by the APS and e) as measured by the SMPS. Yellow shading indicates major dust events (mSDE, $PM_{10} \geq 100~\mu g$ std m$^{-3}$) and orange shading intermediate dust events (iSDE, $50 \leq PM_{10} \leq 100~\mu g$ std m$^{-3}$). Minor dust events ($20 \leq PM_{10} \leq 50~\mu g$ std m$^{-3}$) are not indicated. Green shading indicates a biomass burning event (BB1).





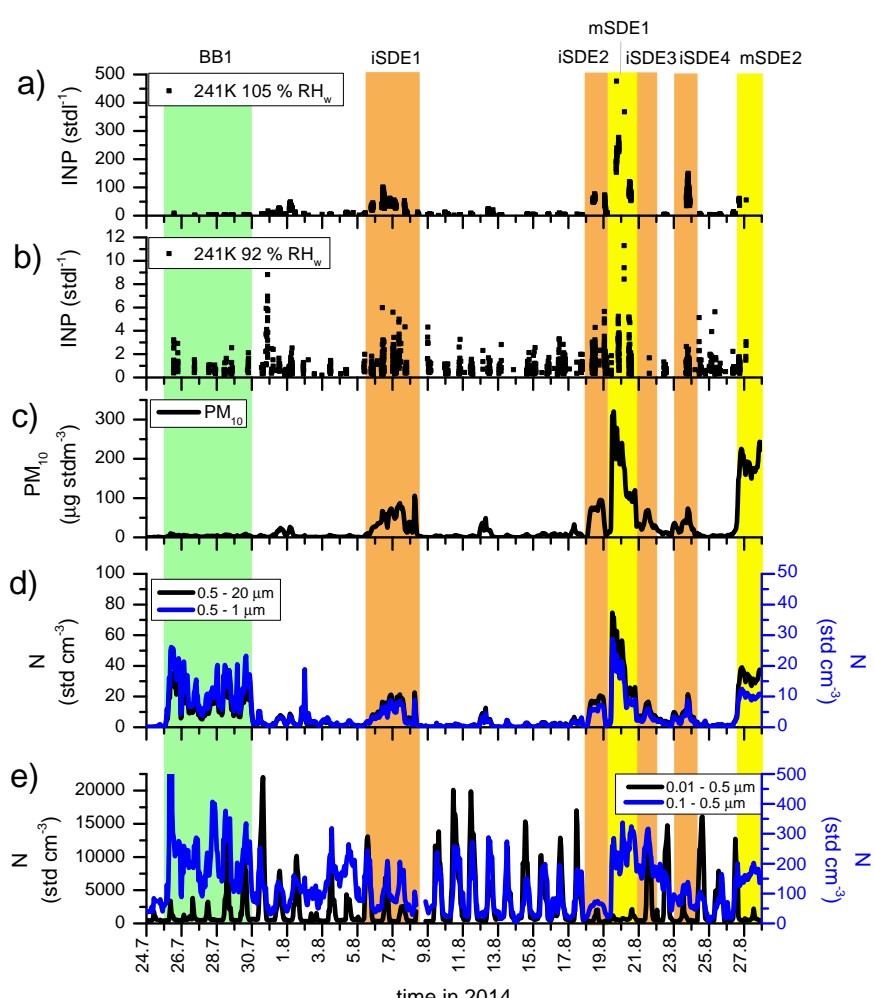

**Figure 3.** As in Fig. 2, but for CALIMA2014.





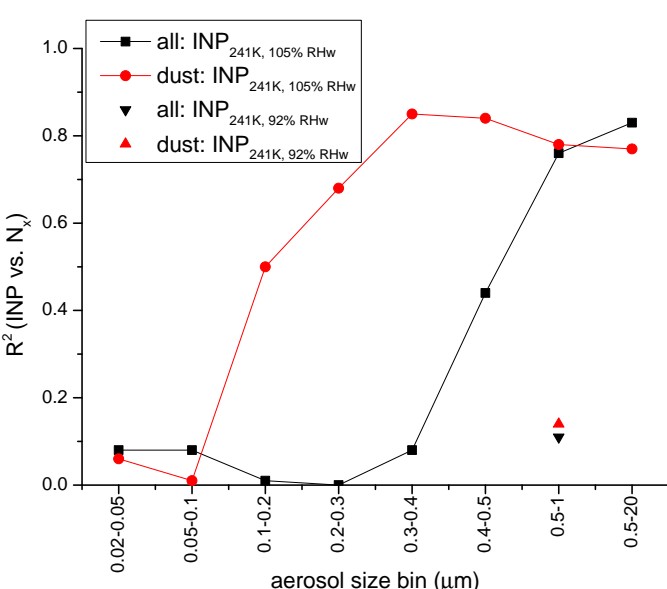

**Figure 4.** Correlation of $\left[INP_{241K,\,105\%\,RH_w}\right]$ and aerosol concentration ($N_x$) of particles of different sizes during CALIMA2014 for dust periods and all periods together. Also shown is the $R^2$ for $\left[INP_{241K,\,92\%\,RH_w}\right]$ with $N_{0.5-1\,\mu m}$.





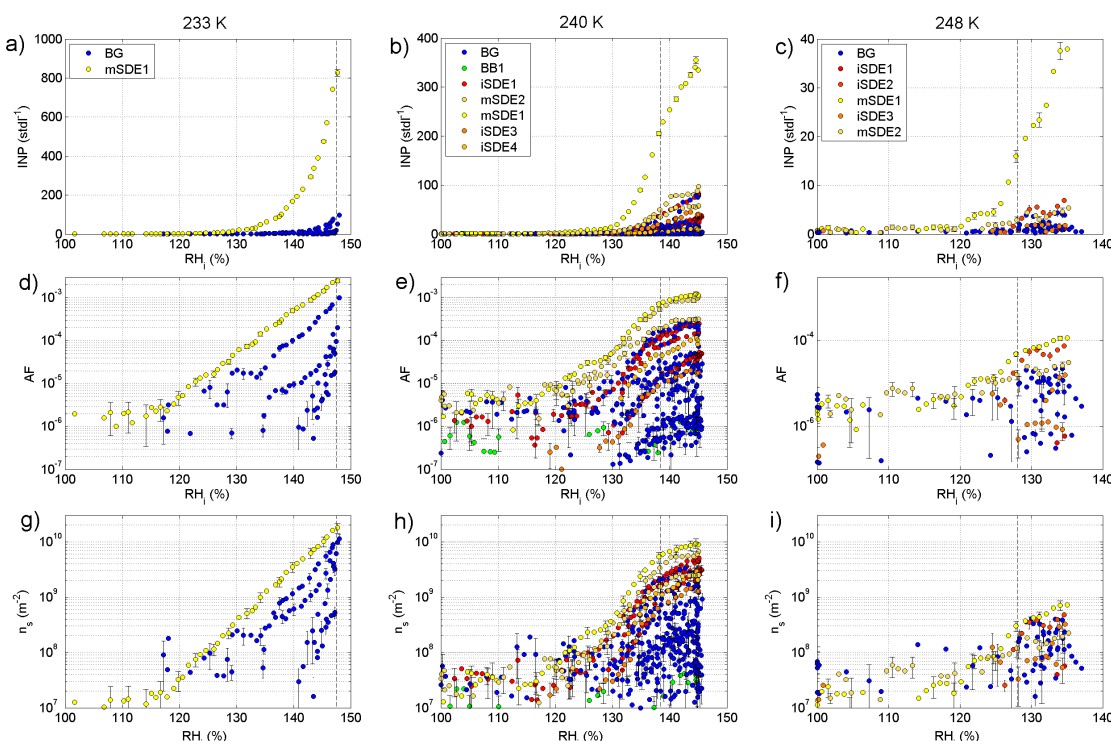

**Figure 5.** $RH_i$ scans of $[INP]$, $AF$ and $n_s$ at 233 K, 240 K and 248 K during CALIMA2014. The dashed vertical lines indicate water saturation. Event types are the same as in Fig. 3 with the addition of BG for background conditions, i.e. not affected by Sahara dust or biomass burning. Error bars are drawn for every third data point.



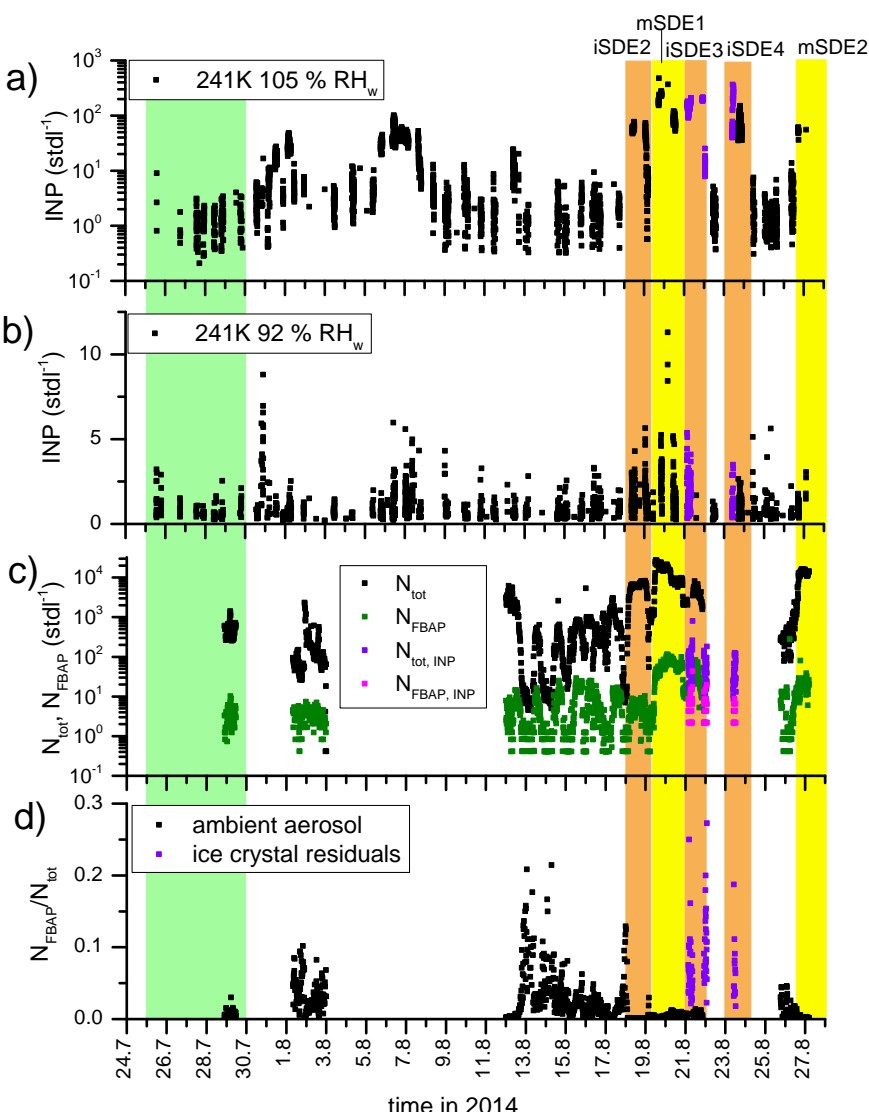

**Figure 6.** $[INP]$ during CALIMA2014 in a) condensation and b) deposition mode at 241 K. Purple data points indicate times when the WIBS was connected downstream of PINC. c) Total and fluorescent particle concentration as measured by the WIBS in parallel (black and green) and in series with PINC (purple and magenta) d) Fluorescent to total particle concentration in parallel (black) and in series (purple) to PINC.





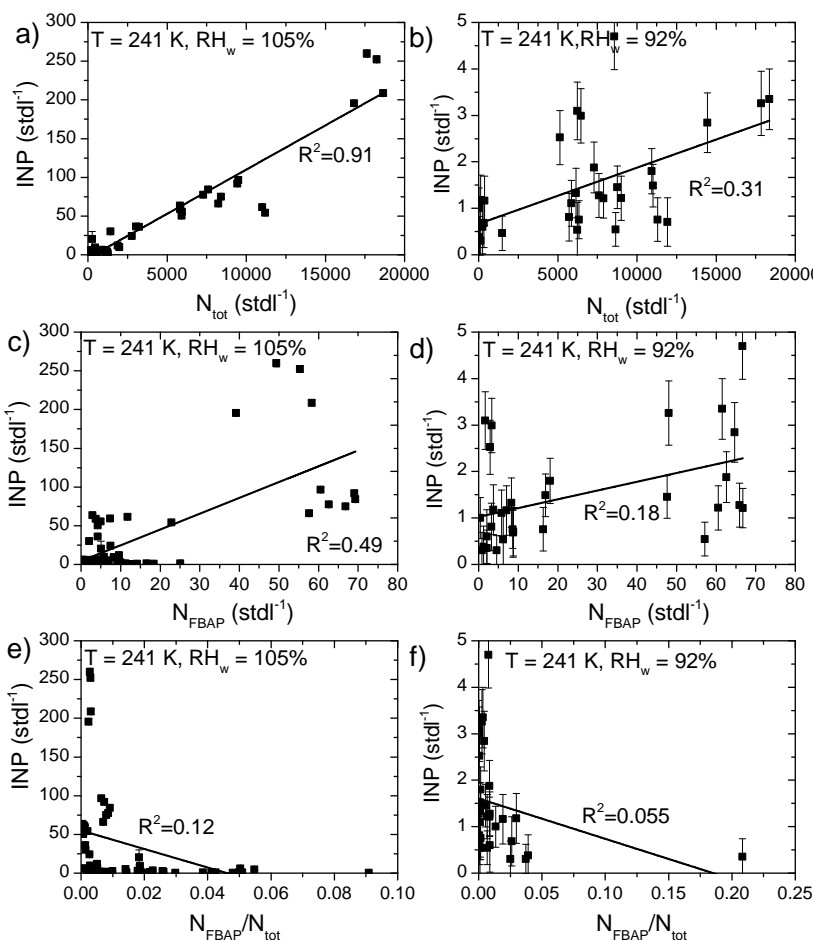

**Figure 7.** Correlation of $[INP]$ at 241 K in a), c) and e) condensation mode and b), d) and f) deposition mode with total and FBAP concentration and the ratio of FBAPs to total particles as measured by the WIBS in parallel to PINC. Error bars are the Poisson statistics based uncertainty.





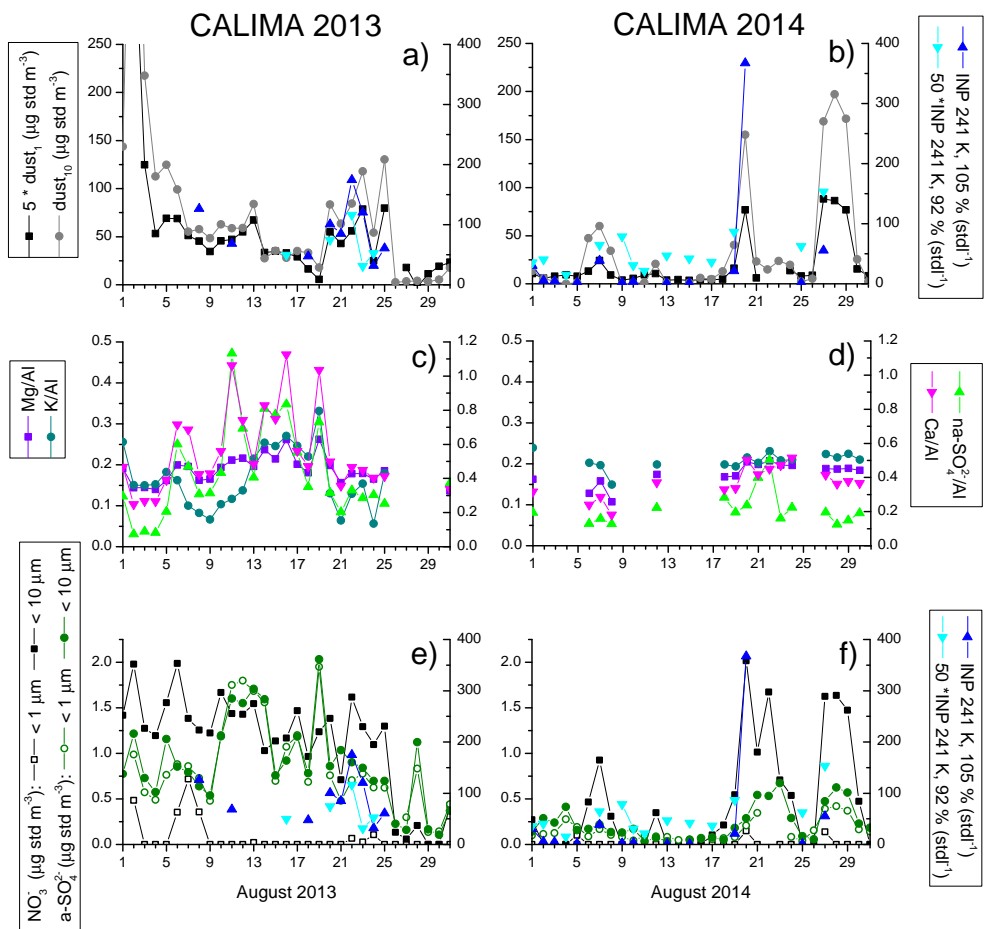

**Figure 8.** Time series of nighttime measurements during CALIMA2013 and CALIMA2014. y-axes labels indicate axes of the respective data in each row.





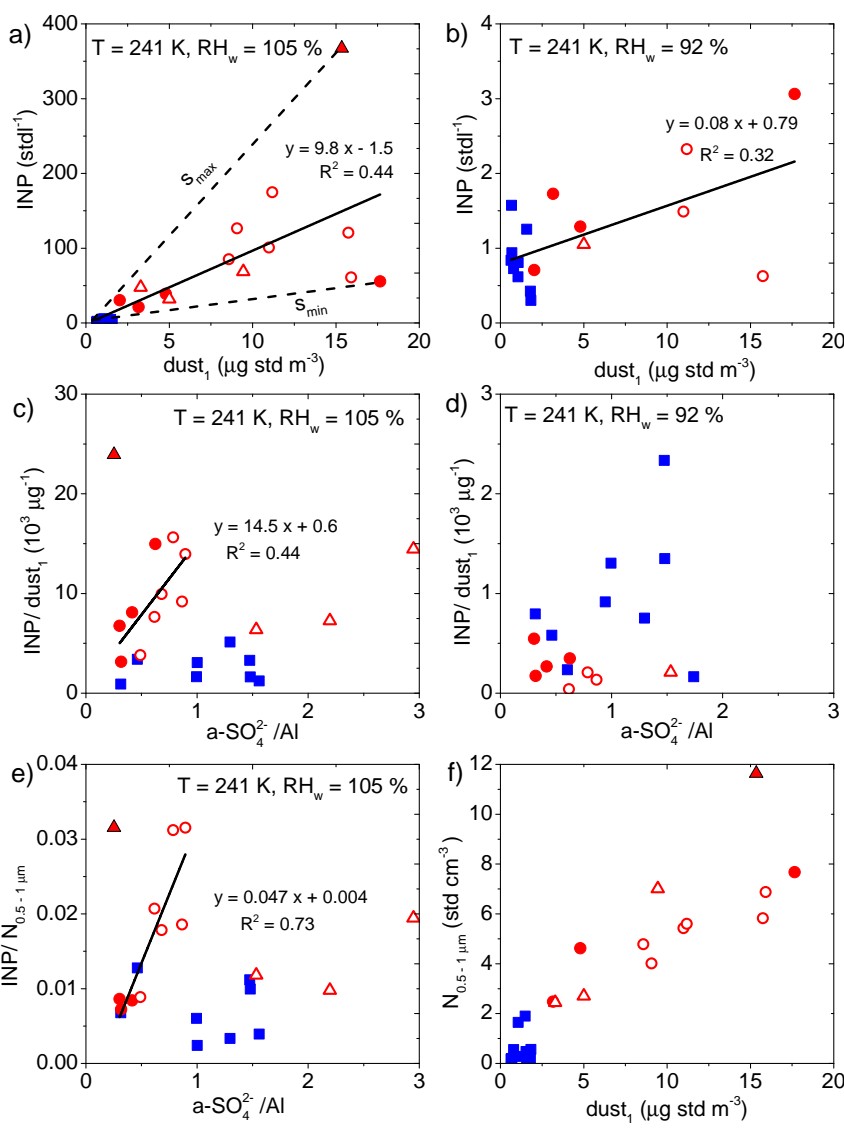

**Figure 9.** a) and b): correlation of $[INP]$ with $dust_1$; c) and d): correlation of $[INP]$/ $dust_1$ with a-$SO_4^{2-}$/ Al; e) correlation of $[INP]$/ $N_{0.5-1\mu m}$ with a-$SO_4^{2-}$/ Al; f) $N_{0.5-1\mu m}$ versus $dust_1$. Plots a), c) and e) show condensation mode $[INP]$ and b) and d) deposition mode $[INP]$ at 241 K. All samples were taken between 22:00-06:00 UTC. Blue squares indicate Atlantic air masses, red circles Sahara influence (open: 2013, filled: 2014) and triangles denote outliers (see text for details).





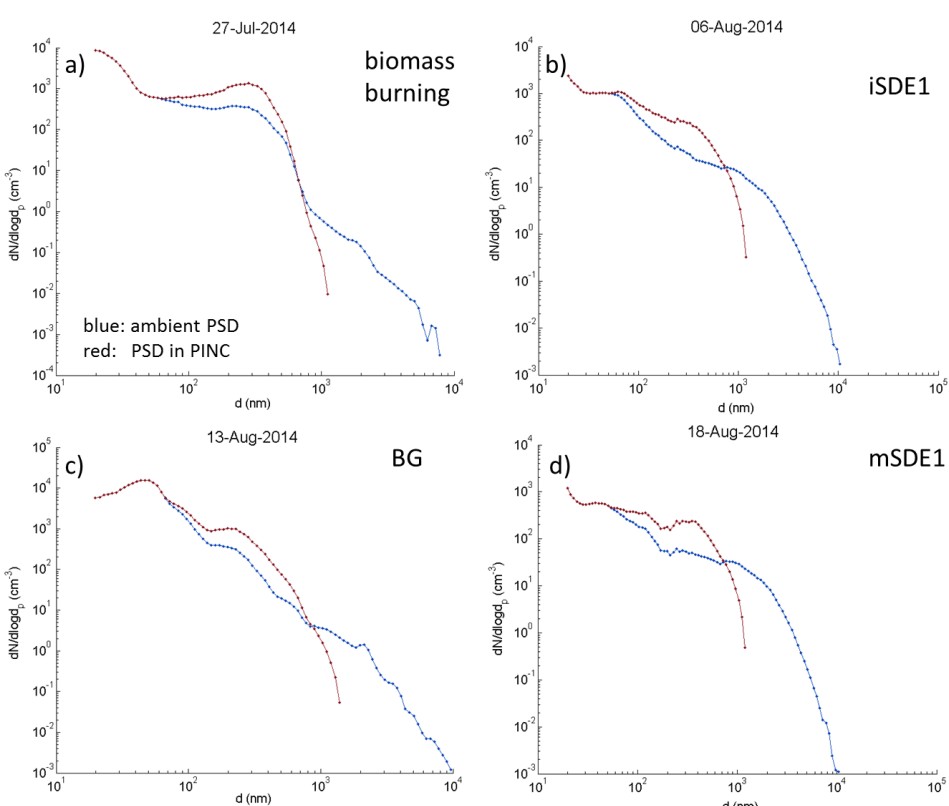

**Figure 10.** Effect of concentrator and impactor on aerosol particle size distribution (PSD) for different event types during CALIMA2014: a) biomass burning event b) intermediate Saharan dust event 1, c) background conditions and d) major dust event 1. In blue are the ambient PSDs, in red the corrected PSDs how they are inside of PINC. Each curve was measured at noon of the respective day.





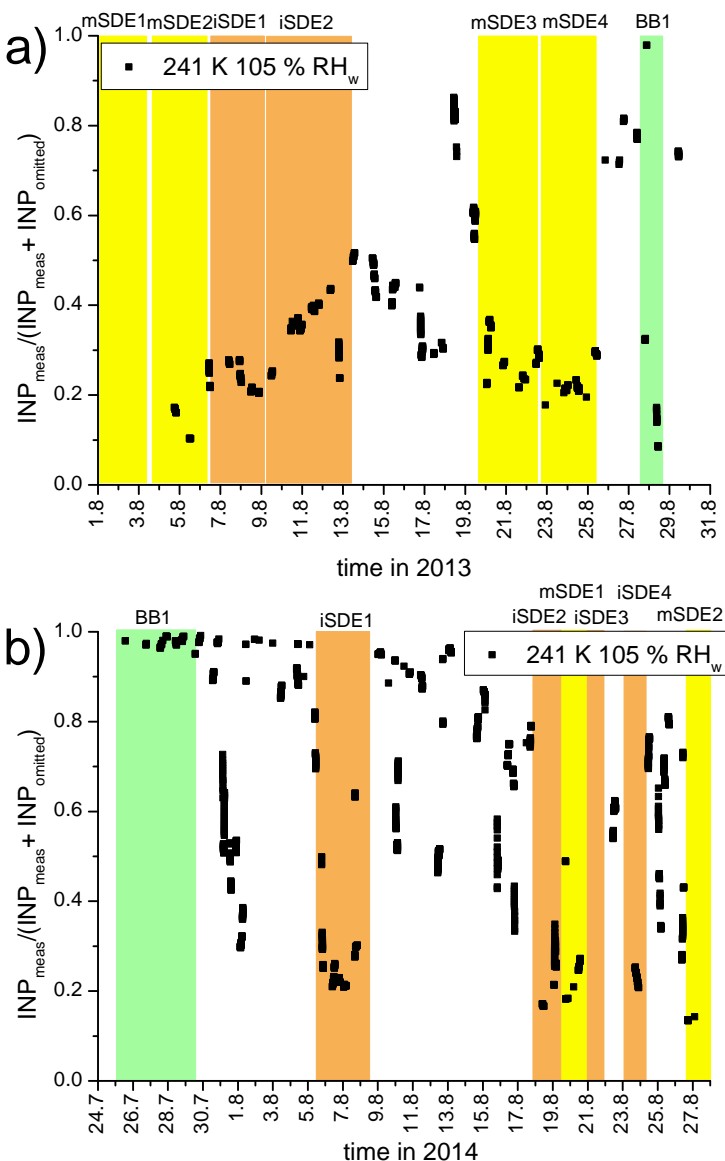

**Figure 11.** a) Time series in 2013 and b) in 2014 of the ratio of measured $[INP]$ to total potential $[INP]$, i.e. the sum of measured $[INP]$ and the calculated omitted $[INP]$ due to the use of the impactor. Color coding refers to events described in Fig. 2 and Fig. 3.





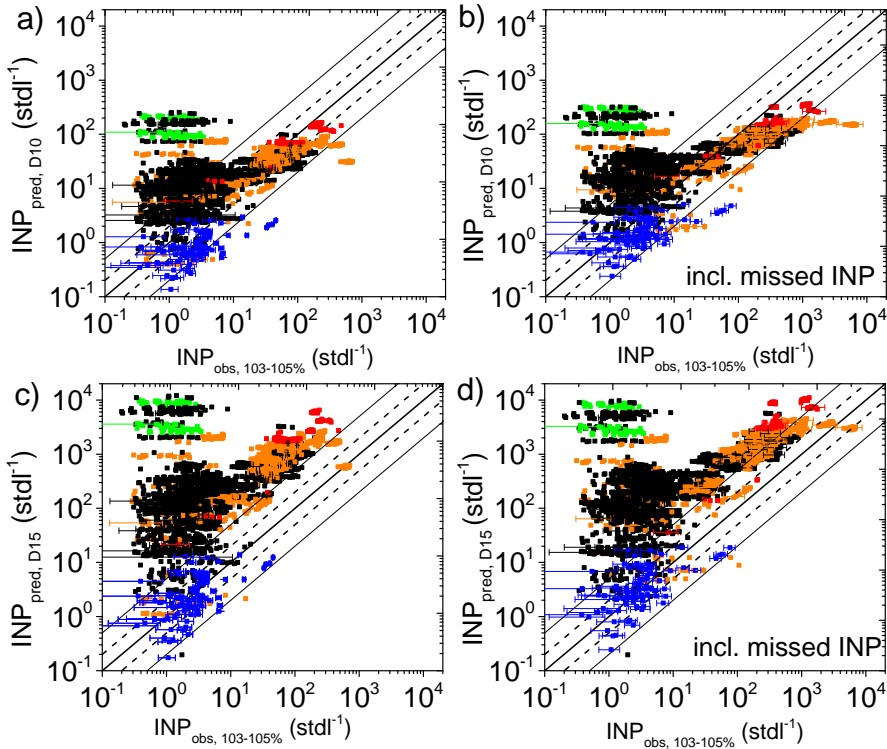

**Figure 12.** Observed $[INP]$ from CALIMA2013 and CALIMA2014 at 105 % $RH_w$ and from CLACE2014 at 103 % vs. predicted $[INP]$ based on the parameterizations from a) and b) DeMott et al. (2010) and c) and d) DeMott et al. (2015). Color coding is as in Fig. 2 with CLACE2014 data shown in blue. The 1:1 line is given as thick solid line, the dashed lines indicate a factor 2 and the thin solid line a factor of 5. The 95 % confidence interval given by DeMott et al. (2015) is about a factor of 4. For the predicted $[INP]$ in a) and c) aerosol particle concentrations corrected for impactor and concentrator were used. For b) and d), the omitted potential INPs were included in the observed $[INP]$ and ambient $N_{0.5\text{-}20\mu m}$ were used for the predicted $[INP]$. Error bars are drawn for 100 random data points per plot. They include the uncertainties of the $[INP]$, $n_s$ and aerosol size distribution measurements.