# Peer review of "Ice nucleating particles in the Saharan Air Layer"

_Atmospheric Chemistry and Physics, 2016_

## Referee Comment (RC1) · Anonymous Referee #1 · 28 Apr 2016

**General comments**

A comprehensive set of data from 2 months of field measurements at Izana, Tenerife is analyzed. The authors have applied advanced experimental methods to measure atmospheric ice nucleating particles (INP), mineral dust, and bioparticles. The subject clearly meets the scope of the journal. These new data that are of special interest, because they characterize relatively young Saharan dust plumes, whereas most data published so far on the ice nucleating properties of mineral dust concern particles that were transported over long distances. From a comparison to the latter the authors plausibly infer that aging and mixture of the dust with pollutants likely will enhance the nucleating properties of mineral dust particles. Bioparticles were found to be slightly enriched among the INP as compared to the total ambient aerosol. In summary, the article is a valuable contribution to our current understanding of the role of mineral dust and bioparticles as INP, on the Saharan dust source, and on the effects of aging. The

manuscript is overall organized and crafted very well, and it deserves publication in ACP with minor revisions only.

Specific comments

Page (P.) 2, line (L.) 16: The reference Klein et al. (2010) may be omitted, because of serious experimental errors in these data, as stated in a recent paper by this laboratory (Schrod et al., Atmos. Meas. Tech., 9, 1313–1324, 2016).

P.2, L. 20: Huang et al., JGR 115, 2010 give a nice climatology of dust AOD over the subtropical atlantic from MODIS, you might add it as a reference here.

P.3, L.20: Conen et al. (2015) measured immersion freezing nuclei, this could be stated.

P.4, L. 19: Uptake of water, deliquescence and growth begin below water saturation. Assuming 100% as a threshold is reasonable for practical reasons, as we usually have nothing better for a given situation, but it is not the truth.

P.7, L.5: I recommend to replace "dried" by "evaporated" or "sublimated". I presume that all ice is completely evaporated once a particle enters WIBS ?

P.7, Chapter 2.5: I wonder if a flow scheme would help a reader who is not familiar with your setup, but I realize that it can be looked up in your JAS paper.

P.8,L.6: Do you have more details or a reference on how gains and losses were accounted for ?

P.10, 3.3, Fig.4 and various other places: when R2 or R are compared, the number N of observations is often useful.

P. 10, L. 5-10: To my knowledge the first one to publish a correlation of INP to the number concentration of "large" particles (0.1-1$\mu$m dp) were Georgii and Kleinjung (Journ. de Recherches Atmosphérique, 145-156, 1967). This reference may be added.

[Figure]

P.13, L.1-2: since you excluded the (blue) squares from the Atlantic sector in your analysis of Fig. 9c you should write "... collected in the SAL under Saharan influence ...". In the same sense you could add, that in Fig. 9d there is no correlation for the red symbols (for red and blue together one might get the impression that there is a correlation).

P.13, last sentence: It is not plausible to use Fig. 9f as an example for the large scattering and variability, because this is the plot that is least scattered of all the 6 plots in Fig.9, less than others that you use for interpretation.

P. 14: I suggest "Potential sampling bias" as a header of chapter 3.7

P. 15, L. 22-23: The parameters that are displayed in Fig.'s 12b and d could be described more clearly . "observed " is misleading, because it is more "what would have been observed, if there were no gains and losses". Also in the sentence "Figure 12b and d show ambient concentrations" the term "ambient" might be understood in this way, one could add, that it is derived from measurements.

P. 16, L29, conclusions: you mention the good correlation of INP to bulk dust mass, but not the much higher correlation of R=0.95 to the total particle number Ntot (Fig. 9a), why ?

P.17, L.5-8, conclusions: you could make a more forceful point of your finding, that ammonium sulfate at the surface of dust increases nucleating properties, by comparing it to the traditional wisdom that insolubility is required for an INP. Pruppacher and Klett (1980) have a whole little chapter 9.2.3.1 named "Insolubility requirement" on that.

Technical corrections

P.11, L.34: Fig. 6e must be 7e.

Fig. 10, caption, last sentence: change to "Each ambient PSD curve was measured ...", because the red PINC PSD curves were calculated, as stated in chapt. 3.7.

[Figure]

P.15, L.19/20 and Fig. 12: Fi. 12 has red symbols, which are described neither in the text nor in captions of Fig. 12 or Fig.2. What is it ?

Fig. 5: the dashed vertical lines indicating water saturation do appear in my printout only for 240 K, but not for 233K and 248K. Maybe the whole Figure can be enlarged ?

---

## Referee Comment (RC2) · Anonymous Referee #2 · 10 May 2016

Boose et al. presented a two-month observational study on ice nucleation particles at the Izana observatory. The INP concentrations between 233-253K were measured with the PINC ice nucleation chamber, together with comprehensive measurements of aerosol properties. They find the increase of ammonium sulfate has a small positive effect on the INP and the biological particle number seems to be higher in INPs than in ambient aerosols. Two widely-used IN parameterizations predict higher number of INPs than the observation. They conclude the current data analysis suggest that the aging process in SAL can lead to an increase ice nucleation efficiency of Sahara mineral dust.

This work is very relevant to the scope of ACP. Such INP measurements are useful for cross validation of existing results from lab experiments and in-situ observations, as well as for model evaluations. The manuscript is clearly written and well organized. Overall I think it is a nice work and I would recommend to publish the paper after some

minor revisions.

Specific comments:

Page 1, Line 14-19: Can you summarize the impact of aging on the deposition nucleation and the condensation nucleation separately? Also, I think it is justified to say the INP measurements and analysis suggest the aging process in SAL can lead to an increase ice nucleation efficiency of Sahara mineral dust, but in my opinion the overestimation of INPs by D10 and D15 (using the observed aerosol properties) does not deliver the same message. Many data used to derive D10 and D15 parameterizations were collected over the Pacific and western/central US, which are far from the Sahara and are more affected by East Asian dust and local dust sources. If the authors indeed want to convey this message (as the current text shows), additional evidences are needed.

Page 3, Line 13: It should be noted that while Sullivan et al. (2010) shows nitric acid can lead to higher ice nucleation rate under supersaturated conditions, it also inhibits the deposition nucleation (sub-saturated).

Page 4, Line 22: Could you please elaborate why RHw= 92% and RHw=105% were chosen for the measurement setup? If a small perturbation was added to it, would the result be sensitive to the change?

Page 5, Line 3: Is the size threshold (>3 micrometer) the only criteria to distinguish ice crystals from droplets? What is the typical size of the droplets measured in PINC?

Page 7, Line 19: Do you mean "analysis"? Reanalysis data are often at coarser resolutions.

Page 10, section 3.3: The analysis and discussion here are very interesting and useful. Would it be useful to calculate the ns function for smaller particles (< 0.5 micro m.) and larger particles separately and compare them?

Page 13, Fig2: Is there a particular reason for using ".8" on the time axis? Would be

nice to use integer numbers. Is the time local time?

Page 14, Line 15: Does this limitation also apply to other types of instruments? In other words, is the poor relationship between INP and the number of >0.5um particles solely because of the instrument limitation? Please comment on this.

Page 29, Fig5: This figure is very informative. I think it is important to mention that the derived ns functions can differ at about one magnitude between various dust events.

Page 36, Fig12 caption, Line 2: "Color coding is as in Fig 2..." What does black color indicate? Background conditions? Please consider adding a legend for convenience of the reader.

Page 36, Fig12 caption: It would be useful to provide a formula showing how the uncertainties were calculated/combined.

---

## Author Comment (AC1) · 28 Jun 2016

We thank Reviewer 1 for their constructive comments. We reproduce reviewer comments in *blue* in the following.

*Page (P.) 2, line (L.) 16: The reference Klein et al. (2010) may be omitted, because of serious experimental errors in these data, as stated in a recent paper by this laboratory (Schrod et al., Atmos. Meas. Tech., 9, 1313–1324, 2016).*

We agree and have deleted this reference.

*P.2, L. 20: Huang et al., JGR 115, 2010 give a nice climatology of dust AOD over the subtropical atlantic from MODIS, you might add it as a reference here.*

We have added the reference.

*P.3, L.20: Conen et al. (2015) measured immersion freezing nuclei, this could be stated.*

We have added "*immersion mode*" before "*INP concentrations at 265 K*" on P.3, L.22 of the revised manuscript.

*P.4, L. 19: Uptake of water, deliquescence and growth begin below water saturation. Assuming 100% as a threshold is reasonable for practical reasons, as we usually have nothing better for a given situation, but it is not the truth.*

We have replaced P. 4, L.19-21

"*Above water saturation condensation freezing, where ice starts forming while water vapor condenses on an INP, as well as immersion freezing, where the INP is immersed in a droplet prior to initiating freezing, were investigated. The latter two processes cannot be distinguished with our method and are thus only referred to as condensation freezing.*"

with P.4, L. 19-22 of the revised manuscript:

"*Close to and above water saturation condensation freezing, where ice starts forming while water vapor condenses on an INP, as well as immersion freezing, where the INP is immersed in a droplet prior to initiating freezing, were investigated. The different processes cannot be distinguished with our method and thus we refer to deposition nucleation at $RH_w < 100$ % and to condensation freezing at $RH_w \geq 100$ %.*"

*P.7, L.5: I recommend to replace "dried" by "evaporated" or "sublimated". I presume that all ice is completely evaporated once a particle enters WIBS?*

Yes, this is the case. We have replaced "*dried*" by "*evaporated*" (now P.7, L.13)

*P.7, Chapter 2.5: I wonder if a flow scheme would help a reader who is not familiar with your setup, but I realize that it can be looked up in your JAS paper."*

We have added a flow scheme for the standard set-up and the coupling of PINC-PCVI and WIBS. This is the new Figure 1 on P.26

P.5 ,L.2-3 : "*A schematic of the experimental set-up is given in Fig. 1.*" was added.

P.7, L. 10: "*An overview of the coupled set-up is given in the right panel of Fig. 1.*" was added.

*P.8,L.6: Do you have more details or a reference on how gains and losses were accounted for ?*

We have added the following sentence on P.8, L. 11-13:

"*As described in the Appendix of  Boose et al. (2016) a size-dependent loss curve of the impactor was measured using montmorillonite and Arizona Test Dust. The size-dependent enrichment of the concentrator was determined using Arizona Test Dust. These loss and gain terms were multiplied with the aerosol particle size distributions.*"

*P.10, 3.3, Fig.4 and various other places: when R2 or R are compared, the number N of observations is often useful.*

We have added the number of observations, "$n_{obs}$"  to all R and $R^2$ values, including those in Table 3.

*P. 10, L. 5-10: To my knowledge the first one to publish a correlation of INP to the number concentration of "large" particles (0.1-1 μm dp) were Georgii and Kleinjung (Journ. de Recherches Atmosphérique, 145-156, 1967). This reference may be added.*

*Georgii and Kleinjung (1967)* was added on p. 10, L. 15.

*P.13, L.1-2: since you excluded the (blue) squares from the Atlantic sector in your analysis of Fig. 9c you should write "… collected in the SAL under Saharan influence…" . In the same sense you could add, that in Fig. 9d there is no correlation for the red symbols (for red and blue together one might get the impression that there is a correlation).*

We have added "*under Saharan influence*" on P. 13, L.18

We have further added "*… for the Saharan samples (red circles in Fig. 10d).*" on p.13, L. 19-20

*P.13, last sentence: It is not plausible to use Fig. 9f as an example for the large scattering and variability, because this is the plot that is least scattered of all the 6 plots in Fig.9, less than others that you use for interpretation.*

We do not intend to use Fig. 9f as an example for large scattering. We agree that the scattering is the lowest here and this makes sense. Actually, if fragmentation took place in a constant manner, one would expect no scattering in this plot at all since with more dust mass the number of dust particles should increase linearly. Therefore, the sentence describes that the fact that there is some scattering is due to the dust agglomerate fragmentation.

To clarify this, we have appended  (P.13, L- 32-35 original manuscript):

"*As shown in Fig. 5h, the surface area alone could not fully explain the differences in observed [INP]. This indicates that the degree of fragmentation of the dust agglomerates (Perlwitz et al., 2015) influences the*

*variability of the number of INP. The scattering of the $N_{0.5-1\,\mu m}$ to dust$_1$ plot (Fig. 10f) illustrates the variability in the dust agglomerates fragmentation."*

as follows (P.14, L.14-18 in the revised manuscript):

*"As shown in Fig. 6h, the surface area alone could not fully explain the differences in observed [INP]. This indicates that the degree of fragmentation of the dust agglomerates (Perlwitz et al., 2015) influences the variability of the number of INP. If fragmentation was constant, a linear relationship between $N_{0.5-1\,\mu m}$ and the dust$_1$ mass would be expected. The scattering of the $N_{0.5-1\,\mu m}$ to dust$_1$ plot (Fig. 10f) illustrates thus the variability in the dust agglomerates fragmentation."*

*P. 14: I suggest "Potential sampling bias" as a header of chapter 3.7*

The header of chapter 3.7 on P.14 has been changed to "*Potential sampling bias*".

*P. 15, L. 22-23: The parameters that are displayed in Fig.'s 12b and d could be described more clearly . "observed " is misleading, because it is more "what would have been observed, if there were no gains and losses". Also in the sentence "Figure 12b and d show ambient concentrations" the term "ambient" might be understood in this way, one could add, that it is derived from measurements.*

We have changed the respective sentences on p. 15, L. 22-24 (original manuscript)

*"Figure 12b and d show ambient concentrations. Here, the observed [INP] include the omitted INPs, as described in the previous section. Thus, for the predicted [INP] the ambient size distribution of particles between $0.5 \leq d_{ve} \leq 20\ \mu m$ was used without further corrections."*

to p. 16, L. 8-13 (in the current manuscript):

*"Figure 13b and d refer to ambient concentrations. The [INP] displayed on the x-axes are those measured with PINC and corrected for the omitted INPs, as described in the previous section. Error bars include the Poisson error of the measured [INP], 10 % uncertainty of the aerosol particle number concentration and 10 % of the aerosol particle size measurements, 20 % uncertainty assumed for the impactor loss curve and a 40 % uncertainty due to the aerosol concentrator curve. For the predicted [INP] the ambient size distribution of particles between $0.5 \leq d_{ve} \leq 20\ \mu m$ was used without further corrections."*

*P. 16, L29, conclusions: you mention the good correlation of INP to bulk dust mass, but not the much higher correlation of R=0.95 to the total particle number Ntot (Fig. 9a), why ?*

We mentioned only the bulk dust mass here with the reasoning to make a fair comparison to the other chemical elements, since the dust and other chemical elements mass was derived from the filter measurements while the particle number from other instruments (SMPS, APS, WIBS). However, we have added now also the correlation with the particle concentration of particles larger than 0.5 μm. We do not use $N_{tot}$ from Table 3 (R = 0.95) because this only refers to particles larger than 0.8 μm (as measured with the WIBS) but instead use the one from Fig. 5 ($R^2 > 0.75$), referring to particles of $d_p \geq 0.5$ μm (from SMPS+APS):

P. 17, L. 17-18 now reads: "*Submicron INP concentrations in the condensation mode at 240 K were observed to correlate well with the concentration of particles larger than 0.5 µm ($R^2 > 0.75$). Furthermore, they correlated fairly well with the bulk dust mass of particles smaller …*"

*P.17, L.5-8, conclusions: you could make a more forceful point of your finding, that ammonium sulfate at the surface of dust increases nucleating properties, by comparing it to the traditional wisdom that insolubility is required for an INP. Pruppacher and Klett (1980) have a whole little chapter 9.2.3.1 named "Insolubility requirement" on that.*

We have added the following sentence on P.17, L. 29-31:

"*The observation in this work that the presence of a soluble salt ion leads to an improved ice nucleation ability of dust particles questions the conventional assumption of insolubility as a requirement for INPs (Pruppacher and Klett, 1997).*"

***Technical corrections***

*P.11, L.34: Fig. 6e must be 7e.*

This has been changed ( P.12, L12 revised manuscript. "*8e*" now because of the new Figure 1).

*Fig. 10, caption, last sentence: change to "Each ambient PSD curve was measured …", because the red PINC PSD curves were calculated, as stated in chapt. 3.7.*

The sentence has been changed to (now caption Fig.11, P.36):

"*Each ambient PSD curve was measured at noon of the respective day.*"

*P.15, L.19/20 and Fig. 12: Fi. 12 has red symbols, which are described neither in the text nor in captions of Fig. 12 or Fig.2. What is it?*

We have replaced the sentence in the caption of Fig. 12:

"*Color coding is as in Fig. 2 with CLACE2014 data shown in blue.* "

to (now Fig.13, P. 38):

"*Green data points refer to biomass burning events, orange and red points to intermediate and major dust events, respectively, and black data points to the remaining time periods. CLACE2014 data are shown in blue.*"

*Fig. 5: the dashed vertical lines indicating water saturation do appear in my printout only for 240 K, but not for 233K and 248K. Maybe the whole Figure can be enlarged ?*

We have enlarged the figure (now P.31) and made the vertical lines thicker. We will keep it in mind for the final version of the paper.

---

## Author Comment (AC2) · 28 Jun 2016

We thank Reviewer 2 for their constructive comments. We reproduce reviewer comments in *blue* in the following.

*Page 1, Line 14-19: Can you summarize the impact of aging on the deposition nucleation and the condensation nucleation separately? Also, I think it is justified to say the INP measurements and analysis suggest the aging process in SAL can lead to an increase ice nucleation efficiency of Sahara mineral dust, but in my opinion the overestimation of INPs by D10 and D15 (using the observed aerosol properties) does not deliver the same message. Many data used to derive D10 and D15 parameterizations were collected over the Pacific and western/central US, which are far from the Sahara and are more affected by East Asian dust and local dust sources. If the authors indeed want to convey this message (as the current text shows), additional evidences are needed.*

We do not intend to suggest that aging processes in the SAL lead to enhanced ice nucleation ability solely by comparing to the D10 and D15 parameterization. We have made changes to the abstract and the conclusion section as listed below to stress that we base the conclusion about the enhancing effect of aging first of all on our observations regarding ammonium sulfate and the biological particles. We acknowledge that the comparison to the D10 and D15 parameterizations is not evidence of aging increasing IN ability but rather alludes to the same point. Particularly the comparison with D15 is valuable in this respect because D15 is based on dust from Asia and the Sahara.

We have replaced P.1, L.14-15:

"*We find that an increase of ammonium sulfate, linked to anthropogenic emissions in upwind distant anthropogenic sources, mixed with the desert dust, has a small positive effect on the INP per dust mass ratio. Furthermore, the relative abundance of biological particles was found to be significantly higher in INPs compared to the ambient aerosol. Two common parameterization schemes for INP concentrations, which were derived mostly from atmospheric measurements far away from the Sahara, were found to predict more INPs based on the aerosol load than we observed in the SAL. Overall, this suggests that atmospheric aging processes in the SAL can lead to an increase in ice nucleation efficiency of mineral dust from the Sahara.*"

with P.1, L.14-21 (new manuscript):

"*We find that an increase of ammonium sulfate, linked to anthropogenic emissions in upwind distant anthropogenic sources, mixed with the desert dust, has a small positive effect on the condensation mode INP per dust mass ratio but no effect on the deposition mode INP. Furthermore, the relative abundance of biological particles was found to be significantly higher in INPs compared to the ambient aerosol. Overall, this suggests that atmospheric aging processes in the SAL can lead to an increase in ice nucleation ability of mineral dust from the Sahara. INP concentrations predicted with two common parameterization schemes, which were derived mostly from atmospheric measurements far away from the Sahara, but influenced by Asian and Saharan dust, were found to be higher based on the aerosol load than we observed in the SAL, further suggesting aging effects of INPs in the SAL.*"

Furthermore we have changed the sentence on P.17, L.13:

*"This could be an indication that atmospheric processing as it occurs during transatlantic or transeuropean advection of dust may enhance the ice nucleation ability of mineral dust compared to that after a relatively short atmospheric transport between the Sahara and Tenerife."*

to (new manuscript) P.18, L.3-7:

*"The enhancing effect of ammonium sulfate on ice nucleation, the higher number of FBAPs in INPs compared to the total ambient aerosol and the comparison particularly to the D15 parameterization could be an indication that atmospheric processing as it occurs during transatlantic or transeuropean advection of dust may enhance the ice nucleation ability of mineral dust compared to that after a relatively short atmospheric transport time between the Sahara and Tenerife."*

*Page 3, Line 13: It should be noted that while Sullivan et al. (2010) shows nitric acid can lead to higher ice nucleation rate under supersaturated conditions, it also inhibits the deposition nucleation (sub-saturated).*

We have changed P.3, L.11-14:

*"Condensation of sulfuric acid (Knopf and Koop, 2006; Sihvonen et al., 2014; Wex et al., 2014) was observed to mostly impair ice nucleation, whereas ammonium (Salam et al., 2007; Koop and Zobrist, 2009), nitric acid (Sullivan et al., 2010), or the exposure to ozone (Kanji et al., 2013) can promote it."*

to P.3, L.11-15:

*"Condensation of sulfuric acid (Knopf and Koop, 2006; Sihvonen et al., 2014; Wex et al., 2014) was observed to mostly impair ice nucleation, whereas ammonium (Salam et al., 2007; Koop and Zobrist, 2009), or the exposure to ozone (Kanji et al., 2013) can promote it. Sullivan et al (2010) observed that nitric acid promoted ice nucleation above water saturation but inhibited deposition nucleation below water saturation."*

*Page 4, Line 22: Could you please elaborate why RHw= 92% and RHw=105% were chosen for the measurement setup? If a small perturbation was added to it, would the result be sensitive to the change?*

We have added the following sentence on P.4, L.23-26 in the revised manuscript to explain the choice of conditions:

*"These conditions were chosen such that a high enough fraction of the dust particles should activate as INP to be measureable with PINC, to be able to clearly distinguish between deposition and condensation/ immersion mode and to compare to an earlier study on free tropospheric INP at the Jungfraujoch in the Swiss Alps conducted under similar conditions."*

*Page 5, Line 3: Is the size threshold (>3 micrometer) the only criteria to distinguish ice crystals from droplets? What is the typical size of the droplets measured in PINC?*

The size of the droplets depends on the $RH_w$ conditions in the chamber as well as the temperature. Indeed under certain conditions the droplets can grow to 3 µm as well. We have added now a part on

the evaporation section of PINC which describes in more detail why the size threshold of 3 μm can be used as sole criterion to distinguish ice crystals from droplets and interstitial aerosol.

We have replaced P.4, L.32 – P5., L.4 in the original manuscript:

"*An impactor with an aerodynamic D50 cut-off diameter of 0.9 μm (diameter at which 50 % of the particles impact) was used upstream of PINC to allow a distinction by size of larger ice crystals which had formed in PINC and unactivated aerosol particles and droplets. Ice crystals, droplets and aerosol particles in the size range 0.5-25 μm were detected with an Optical Particle Counter (OPC; Lighthouse REMOTE 5104; Fremont, USA) downstream of PINC. Particles larger than 3 μm were classified as ice crystals.*"

with P.5, L.3 – 11 in the revised manuscript:

"*In the standard set-up an impactor with an aerodynamic $D_{50}$ cut-off diameter of 0.9 μm (diameter at which 50 % of the particles impact) was used upstream of PINC to allow a distinction by size between larger ice crystals which formed in PINC and unactivated aerosol particles and droplets. In the evaporation section at the lower part of PINC the wall temperatures are kept both at the warm wall's temperature, maintaining $RH_i$ = 100 % while $RH_w$ < 100 %, leading to droplet evaporation while ice crystals are preserved. Ice crystals, droplets and aerosol particles in the size range 0.5-25 μm are detected with an Optical Particle Counter (OPC; Lighthouse REMOTE 5104; Fremont, USA) downstream of PINC. Particles larger than 3 μm are classified as ice crystals. Under high $RH_w$ conditions, droplets may grow to sizes larger than 3 μm and a differentiation by size is not possible anymore. This droplet breakthrough occurs at $RH_w$ = 108 % for our sampling conditions (T = 240 K).*"

*Page 7, Line 19: Do you mean "analysis"? Reanalysis data are often at coarser resolutions.*

Yes, this was wrong. We have corrected it to (P.7, L.25-26 in the revised manuscript):

"*ECMWF analysis data were used as input and the model was run with a resolution of 0.25°.*"

*Page 10, section 3.3: The analysis and discussion here are very interesting and useful. Would it be useful to calculate the ns function for smaller particles (< 0.5 micro m.) and larger particles separately and compare them?*

Unfortunately, we cannot allocate the fractions of INPs smaller than 0.5 μm and larger than 0.5 μm. Thus, it is impossible to calculate $n_s$ for the two different size fractions independently. To answer the reviewers comment to see if there is a relation between $n_s$ and the small particle fraction, we have calculated $n_s$ for the full size range and made the same comparison with the number of aerosol particles in each size bin as we had done in the manuscript with the INP (see below). Please note that we have plotted R instead of $R^2$ here.

The correlation of the INP concentrations with the size bins (as done in the manuscript) warrants the usage of $n_s$ since larger size bins correlate better with the INP concentrations. Correlating now $n_s$ with particles in the different size bins, the clear dependency of INP on size gets lost. This is because $n_s$ ≈INP/$(\pi d^2)$ , hence the denominator and numerator are proportional to $d^x$. We don't know the exact dependency of INP on d but expect x in this case to be close to 2 since ice nucleation is a surface process.

We see now in the plot below that R peaks for aerosol particles of $d_p$ = 0.4-0.5 µm, which are the particles providing the highest fraction to the total surface area during most of the CALIMA campaigns. However, all R values are very low. Since we don't think the plot is helping the reader to better understand the size dependency of ice nucleation, we do not include this plot in the revised manuscript.

[Figure]

*Page 13, Fig2: Is there a particular reason for using ".8" on the time axis? Would be nice to use integer numbers. Is the time local time?*

".8" was referring to August and was missing the "." after "8". We've changed the date format now from "DD.M" to "DD.MM." in Fig. 3, Fig. 4, Fig. 7 and Fig. 12 of the revised manuscript.

*Page 14, Line 15: Does this limitation also apply to other types of instruments? In other words, is the poor relationship between INP and the number of >0.5um particles solely because of the instrument imitation? Please comment on this.*

This limitation is common for CFDC-like instruments which use size as criterion for differentiation between droplets and ice crystals. The impactor's $D_{50}$ of different instruments varies and the one of PINC is rather at the lower end. This is why we provide this discussion. However, the correlation during the CALIMA campaigns between the condensation INP concentrations at 240 K and aerosol particles > 0.5um is not poor ($R^2$ > 0.78). It only is so for the deposition mode INP at 240 K ($R^2$<0.2). The poor correlation with the deposition mode INP likely is due to the low activated fractions at this $RH_w$ which is more than a factor 15 lower than that of the condensation mode INP.

*Page 29, Fig5: This figure is very informative. I think it is important to mention that the derived ns functions can differ at about one magnitude between various dust events.*

We fully agree. We have stated on P.11, L. 1 (original manuscript):

*"In addition, differences between the different SDEs can be found by up to an order of magnitude which must be related to the composition of the aerosol particles."*

and have replaced it with P.11, L.12-13 (new manuscript):

*"In addition, differences of up to one order of magnitude in $n_s$ between the different SDEs are found which must be related to the composition of the aerosol particles."*

*Page 36, Fig12 caption, Line 2: "Color coding is as in Fig 2…" What does black color indicate? Background conditions? Please consider adding a legend for convenience of the reader.*

We have replaced the sentence in the caption of Fig. 12:

*"Color coding is as in Fig. 2 with CLACE2014 data shown in blue. "*

to (now caption of Fig.13, P. 38):

*"Green data points refer to biomass burning events, orange and red points to intermediate and major dust events, respectively, and black data points to the remaining time periods. CLACE2014 data are shown in blue."*

*Page 36, Fig12 caption: It would be useful to provide a formula showing how the uncertainties were calculated/combined.*

The uncertainties have been combined by standard error propagation. However, we realized that we have not yet provided the uncertainties we took into account. This has been added now, in addition to adding an equation on how the INP concentrations including the omitted INPs have been calculated.

Equation 2 on the omitted INPs has been added on P.15, L.9:

*"$INP_{omitted} = n_s\ A_{omitted}$ "*

We have added on P.16, L.8-10 (revised manuscript):

*"Error bars include the Poisson error of the measured [INP], 10 % uncertainty of the aerosol particle number concentration and 10 % of the aerosol particle size measurements, 20 % uncertainty assumed for the impactor loss curve and a 40 % uncertainty due to the aerosol concentrator curve."*